# Heat-activated growth of metastable and length-defined DNA fibers expands traditional polymer assembly

Michael D. Dore [1,7], Muhammad Ghufran Rafique [1,7], Tianxiao Peter Yang[2], Marlo Zorman[3], Casey M. Platnich[1], Pengfei Xu[1], Tuan Trinh [1], Felix J. Rizzuto [4], Gonzalo Cosa [1,5], Jianing Li [6], Alba Guarné [2,5] & Hanadi F. Sleiman [1,5] ✉

Biopolymers such as nucleic acids and proteins exhibit dynamic backbone folding, wherein site-specific intramolecular interactions determine overall structure. Proteins then hierarchically assemble into supramolecular polymers such as microtubules, that are robust yet dynamic, constantly growing or shortening to adjust to cellular needs. The combination of dynamic, energy-driven folding and growth with structural stiffness and length control is difficult to achieve in synthetic polymer self-assembly. Here we show that highly charged, monodisperse DNA-oligomers assemble via seeded growth into length-controlled supramolecular fibers during heating; when the temperature is lowered, these metastable fibers slowly disassemble. Furthermore, the specific molecular structures of oligomers that promote fiber formation contradict the typical theory of block copolymer self-assembly. Efficient curling and packing of the oligomers – or 'curlamers' – determine morphology, rather than hydrophobic to hydrophilic ratio. Addition of a small molecule stabilises the DNA fibers, enabling temporal control of polymer lifetime and underscoring their potential use in nucleic-acid delivery, stimuli-responsive biomaterials, and soft robotics.

The ability of biopolymers such as proteins to fold into well-defined conformations is central to their function[1–4]. Monomer sequence dictates folding, and small changes to chemical or physical properties—such as hydrophobicity or charge—can completely alter the folding pathway, resulting structure, and function of the biomolecule[5]. Folded proteins can further assemble into supramolecular extended structures, such as actin filaments and microtubules, allowing the cell to adapt, move, and grow. These exemplar supramolecular polymers are soft, water-soluble, and dynamic: they require a stimulus to grow, and their length is controlled on demand. Synthetic systems that realize these properties may drive major advances in biotechnologies, ranging from tissue regeneration and biologic manufacturing to rewritable information storage and nanomaterials organization[6–8]. In addition, many natural supramolecular fibers, such as collagen, are entropically controlled—assembly occurs above a floor temperature[9]. Synthetic supramolecular polymers typically melt at high temperatures and form again upon cooling, due to their dependence on specific intermolecular interactions such as hydrogen bonding. Reversing this paradigm would enable the creation of unique supramolecular materials with functions activated by heat.

Inspired by protein folding, the field of foldamers has created synthetic, sequence-defined oligomers that fold into ordered

[1]Department of Chemistry, McGill University, 801 Sherbrooke St W, Montréal, QC H3A 0B8, Canada. [2]Department of Biochemistry and Centre de Recherche en Biologie Structurale, McGill University, Montréal, QC, Canada. [3]Department of Chemistry, University of Vermont, Burlington, VT 05405, USA. [4]School of Chemistry, University of New South Wales, Sydney, NSW 2052, Australia. [5]Centre de Recherche en Biologie Structurale, McGill University, Montréal, QC, Canada. [6]Department of Medicinal Chemistry and Molecular Pharmacology, Purdue University, West Lafayette, IN 47906, USA. [7]These authors contributed equally: Michael D. Dore, Muhammad Ghufran Rafique. ✉e-mail: hanadi.sleiman@mcgill.ca

conformations, leading to numerous applications in biology, sensing, and materials science[10–13]. Emulating the diverse properties of biopolymers using synthetic oligomers requires understanding how subtle changes in monomer sequence impact self-assembly: from the folding of individual oligomers to their hierarchical assembly, and the dynamics of their resultant nanostructures. Like peptides or proteins, nucleic acids may also fold into highly ordered structures, including the folding of RNA into enzymatic ribozymes or of DNA into guanine quadruplexes and i-motifs[14–17]. This behavior is enabled by the anionic phosphodiester backbone and its interplay with ions combined with the hydrophobic character and molecular recognition of nucleobases. Incorporating synthetic units into the structure of nucleic-acid polymers diversifies the toolbox of interactions we can harness to program unique conformational morphologies[18–21]. In advancing our understanding of how monomer order and identity impacts function, we can use artificial units to program the assembly of DNA nanomaterials for delivery and tissue engineering applications, among others[22–25].

A particularly exciting new class of synthetic macromolecules inspired by DNA are sequence-defined poly(phosphodiesters) (SDPPs): polymers with a phosphate backbone and artificial monomers arranged in well-defined sequences that are generated on an automated DNA synthesizer[26–29]. Their emerging applications range from drug delivery and gene silencing to data storage and organic electronics[30–32]. Like DNA, SDPPs are capable of folding into well-defined conformations because of their anionic backbone and generally hydrophobic monomers. SDPPs can be readily appended to natural DNA sequences and these sequence-defined DNA oligomers can self-assemble into spheres, fibers, and sheets in aqueous solution while maintaining the utility of their DNA sequences in self-assembly and gene silencing[33–41]. Developing dynamic soft nanomaterials that can alter their chemical, biological, or physical properties from the bottom-up via specific stimuli paves the way for autonomous, life-like devices inspired by natural systems.

In this work, we report the heat-activated and heat-stabilized seeded growth of length-defined supramolecular polymers, which slowly disassemble once cooled. Previously, we demonstrated a specific sequence-defined DNA oligomer assembled into stable DNA fibers through a mechanism requiring the folding of the poly(phosphodiester) block into a disc-like geometry[42]. Here, we report the discovery that this behavior is highly sequence and length-dependent and show that a new hydrophobic oligomer undergoes heat-activated supramolecular growth into length-defined DNA fibers of low dispersity via a seeded-growth process[9,43]. The final self-assembled morphology cannot be predicted by simple microphase separation rules; instead, both experimental and molecular dynamics simulations show that fiber formation is conditional on the dimensions of a curled conformation of the oligomer, determined by the degree of polymerization and alkyl chain length. The unique high-temperature dependency and stability of the self-assembly also contrast with typical supramolecular fiber systems. Moreover, despite their dynamic exchange and disassembly upon cooling to room temperature, the fiber morphology remains in a metastable state on the timescale of hours; much like natural microtubules, their structure is temporarily maintained once the stimulus is turned off[44]. Using a small molecule that non-covalently crosslinks thymine residues, we show that this dynamic morphology can be "frozen" into stable fibers that retain low dispersity. This work defines a new class of sequence-defined DNA oligomers called 'curlamers', whose folding behavior is precisely dictated by sequence and disc geometry, and whose supramolecular polymerization combines dynamic character with length control—two properties that are often mutually exclusive in synthetic materials[45].

## Results
### High-temperature supramolecular polymerization
The sequence-defined DNA oligomer bC8$_8$-DNAa consists of eight units of a branched alkyl chain-based monomer containing eight

carbons, bC8, appended to a 19mer arbitrary sequence of DNA (Fig. 1A). Initially—at room temperature in aqueous solution (pH 8) with 6.25 mM Mg$^{2+}$—no self-assembly of this DNA oligomer was observed by atomic force microscopy (AFM, Supplementary Fig. 1). When heated incrementally at 0.5 °C per minute, fibers longer than 5 μm were observed above 85 °C (Fig. 1B and Supplementary Figs. 2, 3). The hot solution was drop-cast directly onto mica and immediately washed with water before imaging. Fiber assembly was abrupt, without a gradual increase in fiber length as temperature increased[42]. This sharp transition was also investigated by fluorescence spectroscopy. A cyanine-3 (Cy3) moiety was incorporated in the backbone of the DNA oligomer, between the hydrophilic and hydrophobic blocks (Supplementary Fig. 4). Fluorescence emission steeply decreased as temperature rose above 88 °C and assembly occurred (with a maximum decrease at 92 °C), indicating a change in microscopic environment surrounding the dyes as they became incorporated into fibers (Fig. 1C and Supplementary Figs. 5, 6). As suggested by the AFM data, the fluorescence measurements confirm that self-assembly occurs above a critical temperature, not as the solution cools. A solution of bC8$_8$-Cy3-DNAa held at a constant 90 °C also displayed a rapid decrease in fluorescence intensity, reaching a minimum after just 30 min (Supplementary Fig. 7). In the absence of Mg$^{2+}$, a sharp decrease in fluorescence was not observed, suggesting that divalent cations are required to screen the negative charges of the phosphodiester groups, reducing intermolecular repulsion (Supplementary Fig. 8).

Very slow heating (at 0.1 °C/min) to 75 °C (below the apparent assembly temperature of 88 °C) and holding this temperature for 6 h gave very long and flexible fibers of over 30 μm in length, as observed by total internal reflection fluorescence microscopy (TIRF-M) (Fig. 1D and Supplementary Fig. 9). This assembly at lower temperatures than previously indicated by AFM or fluorescence suggested a high sensitivity to the heating rate. A heat-activated nucleation-growth mechanism could explain these observations, in which the nucleation rate gradually increases with temperature. A slower heating rate could result in a lower assembly temperature by facilitating the growth of only a few nuclei into very long fibers. This would be the opposite of the usual mechanism for supramolecular polymerization, where cooling (rather than heating) rate is used to control the assembly[9,43,46].

### Fiber structure and morphology controlled by DNA oligomer folding
Subtle changes to the length of the bC8 block translated into unexpected changes in self-assembled morphology. Appending two or four additional hydrophobic bC8 monomers to bC8$_8$-DNAa yielded bC8$_{10}$-DNAa and bC8$_{12}$-DNAa. Mirroring the conditions explored in Fig. 1, when slowly heated at 0.5 °C/min from room temperature to 99 °C then quickly cooled, AFM showed medium-length fibers for bC8$_{10}$-DNAa and very short fibers for bC8$_{12}$-DNAa (Fig. 2A and Supplementary Fig. 10). Like bC8$_8$-DNAa, assembly was also followed by fluorescence and a heat-activated transition was again observed (Supplementary Fig. 11). Taken together with bC8$_8$-DNAa, these results showed a transition from very long to very short fibers as the hydrophobic block increased by four monomers from bC8$_8$ to bC8$_{12}$. In contrast, block copolymer self-assembly theory suggests that a larger hydrophobic block decreases the interfacial curvature of the resulting assembly, favouring longer fibers[47]. Slow heating (1 °C/min) to 99 °C followed by slow cooling back to room temperature gave similar results (Supplementary Fig. 12). When quickly heated to 99 °C and slowly cooled at 1 °C/min to room temperature, bC8$_8$-DNAa again assembled into long fibers, bC8$_{10}$-DNAa into very short fibers, and bC8$_{12}$-DNAa into spheres, as observed by AFM and DLS (Supplementary Figs. 13–17). In addition, before any heating, bC8$_8$-, bC8$_{10}$-, and bC8$_{12}$-DNAa all showed no significant assembly by AFM, with only small globular structures observed that may be the individual oligomers (Supplementary Fig. 18). Again, these results indicate that the assembly is

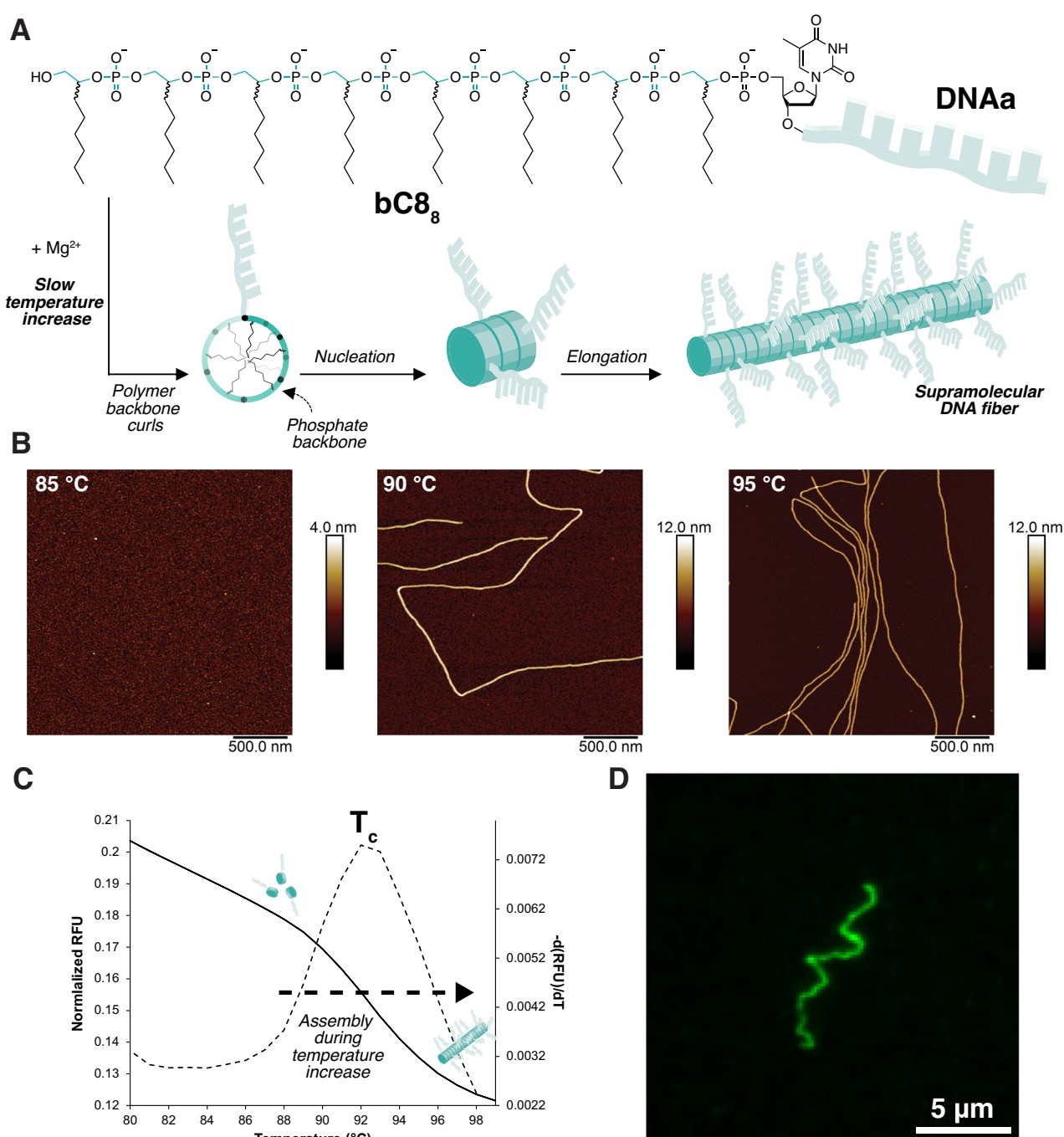

**Fig. 1 | Mechanism of supramolecular fiber assembly of bC8₈-DNAa. A** Molecular structure of bC8₈-DNAa and scheme illustrating its proposed self-assembly mechanism into supramolecular fibers as the temperature increases. **B** AFM images (in air on mica) of bC8₈-DNAa at different temperatures during heating.

**C** Normalized fluorescence of bC8₈-Cy3-DNAa during incremental heating from 20 to 99 °C. **D** TIRF-M image of bC8₈-Cy3-DNAa after slow heating to 75 °C and holding for 6 h.

determined by the heating rate rather than the cooling rate, contrasting with enthalpically controlled supramolecular systems, in which slow cooling facilitates reaching the thermodynamic minimum[9].

Previously, we showed that a similar DNA oligomer (bC12₁₂-DNAa) formed supramolecular fibers[37,42]. Two additional DNA oligomers were synthesized using this bC12 monomer: bC12₈-DNAa and bC12₁₀-DNAa, and assembled by heating rapidly to 95 °C and cooling 1 °C/min to room temperature. In contrast to bC12₁₂-DNAa, bC12₈-DNAa formed spherical assemblies rather than long fibers, and bC12₁₀-DNAa gave shorter fibers (Supplementary Figs. 19–22). This trend from fibers to spheres as the number of bC12 monomers decreased from 12 to 8 is the

opposite of that observed for bC8, where fiber length decreased as the number of bC8 monomers increased. Evidently, the hydrophobic to hydrophilic ratio alone is not governing the assembly morphology.

To characterize their dimensions, we imaged the bC8₈-DNAa and bC8₁₀-DNAa fibers—prepared by slow heating at 0.5 °C/min from room temperature to 99 °C—by cryogenic electron microscopy (cryo-EM) (Fig. 2B, C). These structures were chosen due to their ability to form fibers that would be suitable objects for analysis. For both samples, clearly visible cores with relatively high densities were observed, suggesting densely packed fiber cores. The core diameters of 1.3 ± 0.11 and 1.44 ± 0.09 nm for the bC8₈-DNAa and bC8₁₀-DNAa fibers, respectively,

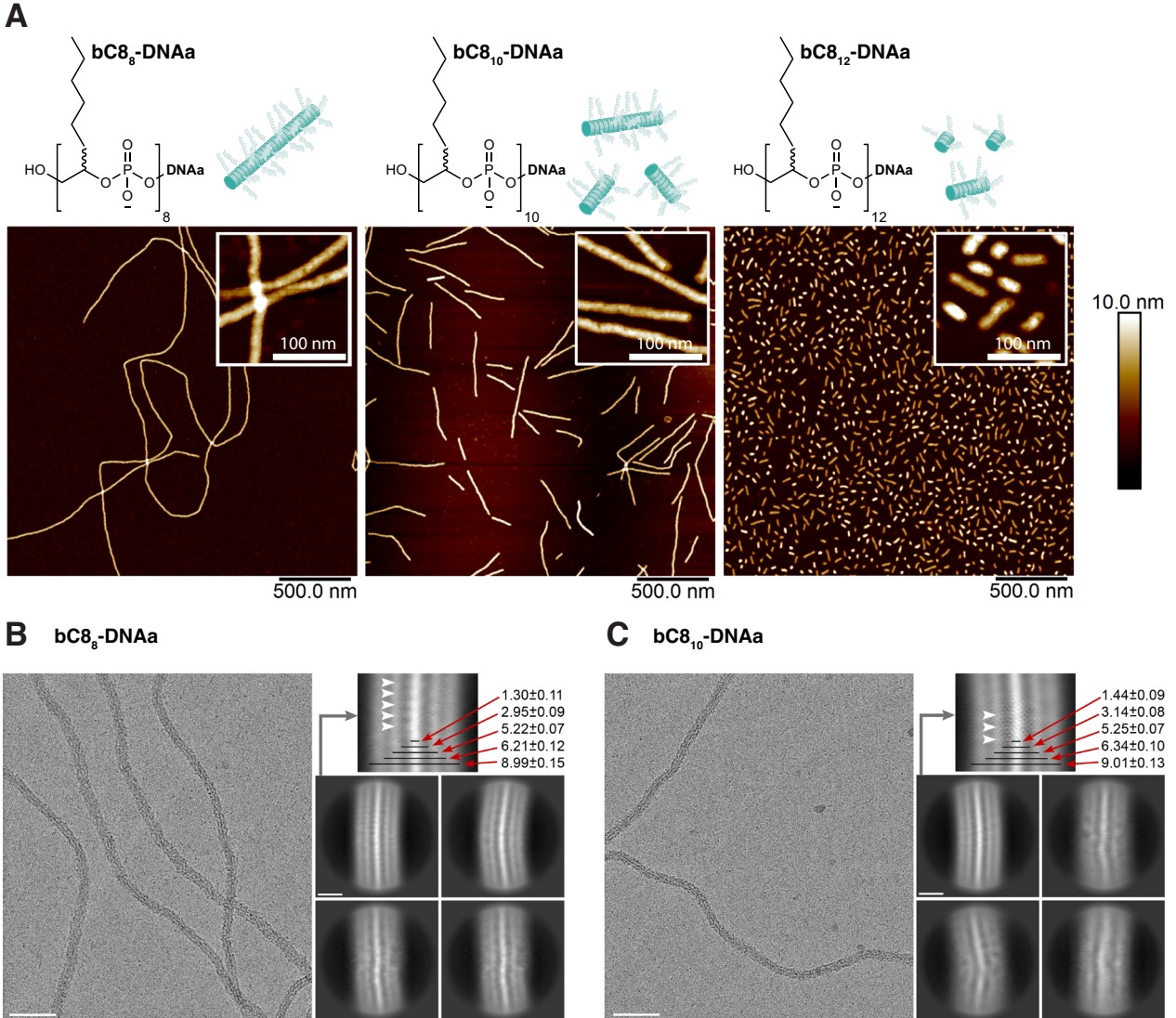

**Fig. 2 | Fiber formation is dictated by oligomer conformation. A** bC8₁₀-DNAa and bC8₁₂-DNAa form short fibers and very short fibers, respectively, in contrast to the long fibers formed from bC8₈-DNAa. AFM images in the air on mica. **B**, **C** Cryo-EM visualization of the bC8₈-DNAa and bC8₁₀-DNAa fibers. Representative cryo-EM micrographs and 2D classes of the bC8₈-DNAa fibers (**B**) and the bC8₁₀-DNAa fibers (**C**). The major 2D classes were used to measure the diameter of the fibers and their core. The dimensions of the fibers are indicated, and white arrowheads mark the wisps protruding from the central core. Scale bars are 500 and 5 nm, respectively.

are less than the lengths of the extended bC8$_N$ blocks (~5 to 6 nm), indicating that the branched, hydrophobic blocks are folded rather than extended. A visible, less dense region (~2.95 nm diameter) flanking the core provides evidence for a more hydrated environment immediately surrounding the alkyl chains. This supports a hypothesis that the core is curling into a more compact conformation in which the alkyl chains are located in the core of the fiber, and the phosphate backbone is wrapped around them[42].

All-atom molecular dynamics (MD) simulations of bC8₈-DNAa and bC8₁₀-DNAa further supported the existence of a curled conformation and indicated an enhanced prevalence at high temperatures. Folding stability was characterized by calculating the relative solvent-accessible surface area (SASA) of the hydrophobic blocks (Supplementary Fig. 23)[48]. At room temperature, bC8₈-DNAa folded into a non-curled, stable, globular conformation, where the alkyl chains of the hydrophobic block aligned perpendicular to the backbone (Fig. 3A, Supplementary Fig. 23, and Supplementary Movies 1–3). At 85 °C, bC8₈-DNAa freely converted between the unfolded state and a structured disc-like, curled one, wherein the backbone curled with alkyl chains pointing inward to the center of the disc (Fig. 3A and

Supplementary S24). These results indicate that high temperatures are required to drive bC8₈-DNAa to fold into a curled conformation, promoting fiber growth. Simulations of bC8₁₀-DNAa at room temperature revealed that it also did not fold into any consistently structured conformation (Supplementary Fig. 23 and Supplementary Movies 1–3). Thus, both cryo-EM and MD simulations support the curling of the core phosphate backbone of bC8₈-DNAa into a disc-like structure at high temperatures that exposes the phosphates to the aqueous solution and directs the alkyl chains to the interior.

The core diameter (ca. 1.3 nm) measured by cryo-EM is smaller than anticipated, if one assumes a perfect disc-like conformation with a radius equal to the length of the branch of the bC8 monomer (7.5 Å, resulting in a diameter of 1.5 nm, Fig. 3B). Instead, we reasoned that the alkyl cores may fold into slightly helical rather than disc-like arrangements, akin to helical foldamers (Fig. 3C). In support of this idea, cryo-EM shows that the bC8₈-DNAa fibers have wisps protruding from the central core that appear to follow a helical symmetry with a pitch of 1.24 ± 0.11 nm (Fig. 2B white arrows). These wisps are also present in the bC8₁₀-DNAa fibers with a similar pitch, but they do not persist along the filament for this oligomer. This suggests that the bC8₈-DNAa

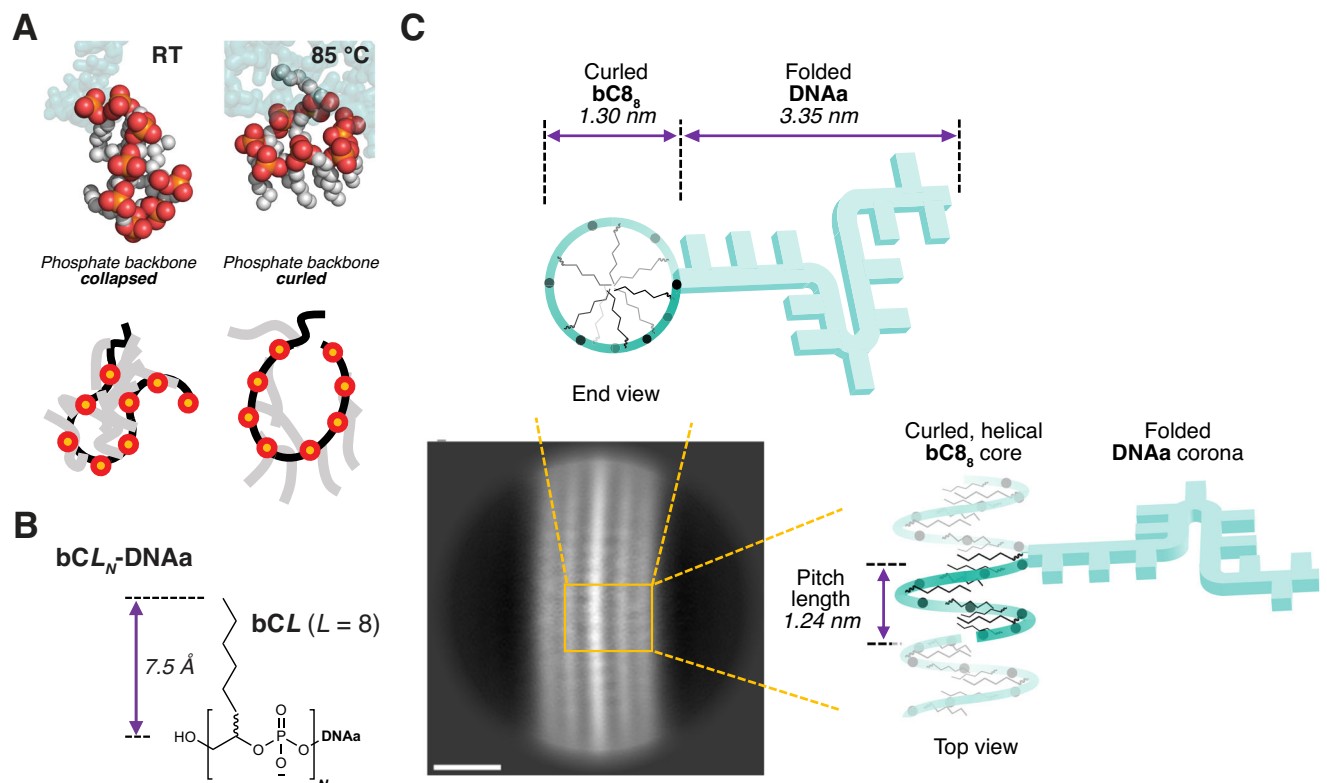

**Fig. 3 | Internal structure and dimensions of DNA fibers. A** Molecular dynamics simulations show that a single bC8$_8$-DNAa oligomer assumes a more structured disc-like conformation at 85 °C compared to room temperature (RT) (images at 300 ns). Phosphates showed in red and orange, the backbone in black, and the alkyl chains in gray. **B** Structure of this class of DNA oligomers with parameters *L* (the length of the monomer's alkyl chain) and *N* (the number of monomers) marked. **C** Proposed curled oligomer conformation and fiber structure dimensions based on the cryo-EM analysis. The blue line with black dots represents the polyphosphodiester backbone of the oligomer. The slightly helical, curled conformation of the bC8$_8$ block is shown, as is the packed DNA block. Scale bar: 5 nm.

oligomer assembles into supramolecular structures with a higher degree of order than the bC8$_{10}$-DNAa oligomer. To maintain a similar helical pitch length as bC8$_8$-DNAa, the longer bC8$_{10}$-DNAa must have a larger core diameter, which was confirmed by both cryo-EM and AFM (Supplementary Figs. 24, 25 and Supplementary Table 1). The larger core diameter for bC8$_{10}$-DNAa combined with the required pitch length may now be a less ideal geometry for the alkyl chains to efficiently occupy the internal space, which could lead to lower fiber propensity.

The bC8$_8$-DNAa fibers had an overall diameter of 8.99 ± 0.15 nm, while the bC8$_{10}$-DNAa fibers measured 9.01 ± 0.13 nm (Fig. 2B, C). These diameters are smaller than expected based on the lengths of the alkyl and DNA chains of each monomer (a DNA 19mer is ~6.5 nm long assuming a helical conformation, and 14 nm if extended, while bC8$_8$ is ~5–6 nm). AFM gave heights even lower than the diameters observed by cryo-EM—however, these are further impacted by the drying of the assemblies (Supplementary Tables 1, 2). Interestingly, by cryo-EM, both fibers showed longitudinal striations corresponding to areas where the fibers are more densely packed. The radial periodicity results in a pattern of alternating areas of higher and lower density, indicating that the DNA chains are not fully extended (Figs. 2B, C, 3C). While we cannot identify the types of interactions that drive DNA compaction, these features explain why the width of the fibers is smaller than anticipated. They also emphasize the importance of the packing of the DNA block to the overall symmetry and, hence, the length of the fibers. Based on the findings of the cryo-EM studies, a proposed structure for the bC8$_8$-DNAa fibers is presented in Fig. 3C.

The efficient helical packing of the oligomeric blocks and therefore their propensity to form long fibers depend on basic structural parameters: the number of monomers (*N*), and the length of the monomer's alkyl chains (*L*) (Fig. 3B). These two parameters define the dimensions of the helical core. Defining the general structure as bC*L$_N$*-DNA, the values of *L/N* for different DNA oligomers were calculated. Interestingly, values closest to 1 correspond to the longest fibers (Supplementary Tables 1, 2). Values above or below 1 gave shorter fibers or spherical structures. For example, *L/N* = 0.8 for bC8$_{10}$-DNAa, implying the number of monomers (*N* = 10) per oligomer is too long, requiring a diameter for a given pitch length that is too wide for the alkyl chains to efficiently fill.

This high dependence on folding geometry highlights how the behavior of these DNA oligomers resembles proteins rather than traditional block copolymer self-assembly. Natural protein supramolecular fiber systems have been identified in which monomeric units must undergo a conformational change to drive nucleation and growth[49]. The system demonstrated here requires heat to trigger a conformational change in which the oligomeric backbone curls into an active state. The propensity of this curled, active state to undergo supramolecular elongation depends on the precise dimensions of the core block: a 1:1 ratio of the number of monomers to alkyl chain length demonstrates the greatest efficiency in fiber elongation (bC8$_8$-DNAa and bC12$_{12}$-DNAa). Our sequence-defined DNA oligomers thus provide a unique opportunity to explore the dramatic morphological transformations that are achievable by altering oligomers one monomer at a time.

**Seeded growth**

As mentioned above, the sudden appearance of long fibers at high temperatures suggests a polymerization mechanism wherein heat activates fiber nucleation and rapid elongation[9]. In seeded-growth systems, control over the number of nuclei allows control over the

number of fibers, like covalent living polymerization[50,51]. Due to their very similar chemical structures, we hypothesized that the short cylindrical particles formed by the fast heating and slow cooling of bC8$_{10}$-DNAa could be used as seeds for the supramolecular polymerization of bC8$_8$-DNAb (where a and b are different random DNA sequences). Because heating is required to drive the elongation of bC8$_8$-DNAb fibers, pre-assembled bC8$_{10}$-DNAa seeds could be added to unassembled bC8$_8$-DNAb oligomers. Subsequent heating would then drive the elongation of bC8$_8$-DNAb fibers from bC8$_{10}$-DNAa seeds at high temperatures.

Pre-assembled, short cylindrical particles of bC8$_{10}$-DNAa were added to a solution of bC8$_8$-Cy3-DNAb at a molar ratio of 1 to 10, then heated at 0.5 °C per minute from 25 to 99 °C while monitoring fluorescence. A transition was observed between 70 and 90 °C, lower than the transition observed for bC8$_8$-Cy3-DNAa alone (Supplementary Fig. 26). In nucleation-growth systems, the energy barrier to self-assembly is typically nucleation rather than elongation; therefore, the lower transition temperature suggested that bC8$_{10}$-DNAa could be acting as seeds to nucleate the assembly of bC8$_8$-Cy3-DNAb. Seeds of bC8$_{10}$-DNAa were again added to bC8$_8$-Cy3-DNAb at a ratio of 1 to 20 and then directly heated at 70 °C (Fig. 4A). An elongation temperature of 70 °C–the very lower edge of the transition observed by fluorescence–was chosen to avoid self-nucleation of bC8$_8$-DNAb. Indeed, nanofibers were observed with very low dispersity in length and grew progressively longer for up to 60 mins, as indicated by AFM (Fig. 4B, C and Supplementary Figs. 27, 28). The observed progressive elongation following longer hold times at 70 °C again indicated the unique ability of this system to actively assemble at high temperatures. The resultant fibers exhibited low dispersity in length ($L_N = 790$ nm, $Đ = 1.02$, $N = 107$–observed by AFM), consistent with a seeded-growth mechanism of polymerization.

By varying the ratio of the nuclei forming bC8$_{10}$-DNAa to the fiber-forming bC8$_8$-DNAb, the length of the fibers could be controlled (Fig. 4D): fibers with lengths ranging from 490 nm ($Đ = 1.07$) to 1730 nm ($Đ = 1.12$) were generated using bC8$_{10}$:bC8$_8$ ratios of 1:10 to 1:40, respectively (Fig. 4D, E). The increase in fiber length with amphiphile ratio was approximately linear, with an average increase in length of $44 \pm 1$ nm per molar equivalent of bC8$_8$-DNAb added (Fig. 4D). Dispersity was observed to slightly increase as higher equivalents of bC8$_8$-DNAb were added, largely due to the presence of shorter-than-average rather than longer fibers (Fig. 4D, E). This may be a result of fibers breaking during AFM sample preparation or a minor degree of uncontrolled nucleation. Another possibility for the shorter fibers is an inherent instability of the assemblies once cooled to room temperature–a dynamic process that we were interested in exploring further.

## Fiber dynamics

Despite their controlled seeded growth, the length-defined DNA fibers were found to be highly dynamic following assembly. After initial seeded growth, fibers were left in solution at room temperature for 5 days: a decrease in number average length from 413 nm ($N = 169$) to 74 nm ($N = 276$) and an increase in dispersity from 1.09 to 1.6 were observed as fibers disassembled (Fig. 5A, B). Reannealing the sample resulted in fiber elongation to 256 nm ($N = 414$) with a dispersity of 1.17. This dispersity is less than what is observed for fibers in the absence of nuclei ($Đ = 1.73$) and much less than what would be expected for a step-growth process ($Đ \sim 2$), suggesting that the bC8$_{10}$-DNAa nuclei may remain present and again act as seeds for the second round of elongation[52]; bC8$_{10}$-DNAa nuclei were in fact observed to be stable at 10 °C for weeks (Supplementary Fig. 29). The regeneration of our nucleated fibers resembles biological systems wherein stimuli drive changes in nanoscale morphology to modulate mechanical forces or compartmentalize processes[53,54]. The heat-activated polymerization demonstrated here could potentially find use in dynamic hydrogels, as

a supramolecular analogue to thermoresponsive covalent polymers such as poly(N-isopropylacrylamide)[55]. Changes in temperature may cause a reversible change in the mechanical properties of a gel, providing a means for actuation or cargo capture and release[56].

The dynamics were also confirmed using fluorescence spectroscopy. In addition to the previously described cyanine-3-labeled DNA oligomer bC8$_8$-Cy3-DNAa, a cyanine-5-labeled DNA oligomer was synthesized: bC8$_8$-Cy5-DNAa. Populations of seeded length-defined Cy3- and Cy5-labeled DNA fibers were separately pre-assembled before being mixed at room temperature and their fluorescence spectra monitored (Fig. 5C). Over the course of 24 h, the Förster Resonance Energy Transfer (FRET) between Cy3 and Cy5 dyes was observed to increase, suggesting increased proximity of the dyes at room temperature (Fig. 5D, E–green circles). AFM revealed that the fibers also disassembled to a large extent, with the lengths decreasing from $L_N = 600$ nm ($Đ = 1.04$, $N = 76$) to $L_N = 165$ nm ($Đ = 1.37$, $N = 155$) (Supplementary Fig. 30). A positive control in which bC8$_8$-Cy3-DNAa and bC8$_8$-Cy5-DNAa were annealed together to form mixed fibers showed a decrease in FRET efficiency from 0.99 to 0.55 over 24 hours (Fig. 5C, E–black circles), and a similar decrease in length from $L_N = 591$ nm ($Đ = 1.07$, $N = 146$) to $L_N = 149$ nm ($Đ = 1.56$, $N = 151$) (Supplementary Fig. 31). The final FRET efficiency value and fiber lengths of the pre-mixed and separately assembled samples are equivalent, indicating equivalent populations regardless of the initial state.

The increase in FRET efficiency over time for the separately assembled **Cy3**- and **Cy5**-fibers suggests that a proportion of mixed assemblies are present that contain both dyes. AFM images showing a highly heterogenous mixture of different-sized particles support the theory that short fibers and small globular aggregates are indeed present following aging at room temperature (resulting in the modest FRET observed), and are likely in coexistence with individual, completely solubilized DNA oligomers that would not be expected to undergo FRET (Fig. 5A). Also, although fiber growth stops after 60 minutes of heating at 70 °C, dispersity increases from this point onwards, providing evidence for fiber dynamics occurring at 70 °C as well (Supplementary Figs. 27, 28).

The addition of the small molecule melamine to nucleated bC8$_8$-DNAa fibers inhibited their disassembly, demonstrating the ability to tune fiber dynamics. Melamine has three faces, two of which have been shown to form complementary hydrogen bonds with thymine in DNA[57]. In the case of a poly(thymine) sequence, this results in the poly(T) being brought together by melamine units[58]. The sequence of bC8$_8$-DNAa contains five consecutive thymine monomers on the 5′ end of the DNAa block, immediately adjacent to the bC8$_8$ block (Fig. 6A). We hypothesized that in this highly DNA-dense environment, melamine may form a network of intermolecular hydrogen bonding interactions between thymine in neighboring bC8$_8$-DNAa oligomers, stabilizing the fibers and preventing disassembly. The use of melamine for this purpose is advantageous over a complementary DNA sequence due to its ability to interact with DNA through two faces at the same time. In combination with melamine's small size and lack of bulky poly(phosphodiester) backbone, this gives the system greater degrees of freedom to form a hydrogen bonding network. Using AFM to image and measure the assemblies, the fiber length ($L_N$) decreased by 75% over 24 h in the absence of melamine, and $Đ$ increased from 1.08 to 1.37 (Fig. 6C and Supplementary Fig. 32). When melamine was added to fibers immediately following assembly, a much smaller $L_N$ decrease of 26% was observed and $Đ$ only increased slightly from 1.08 to 1.14 (Fig. 6B, C). These data support the hypothesis that the rate of disassembly is decreased in the presence of melamine, likely due to the specific hydrogen bonding interactions that melamine forms with thymine. Circular dichroism (CD) indicated that the addition of melamine altered the secondary structure of the DNA within the fibers, as inferred from the increase in ellipticity at 270 nm that has been observed in other DNA-melamine

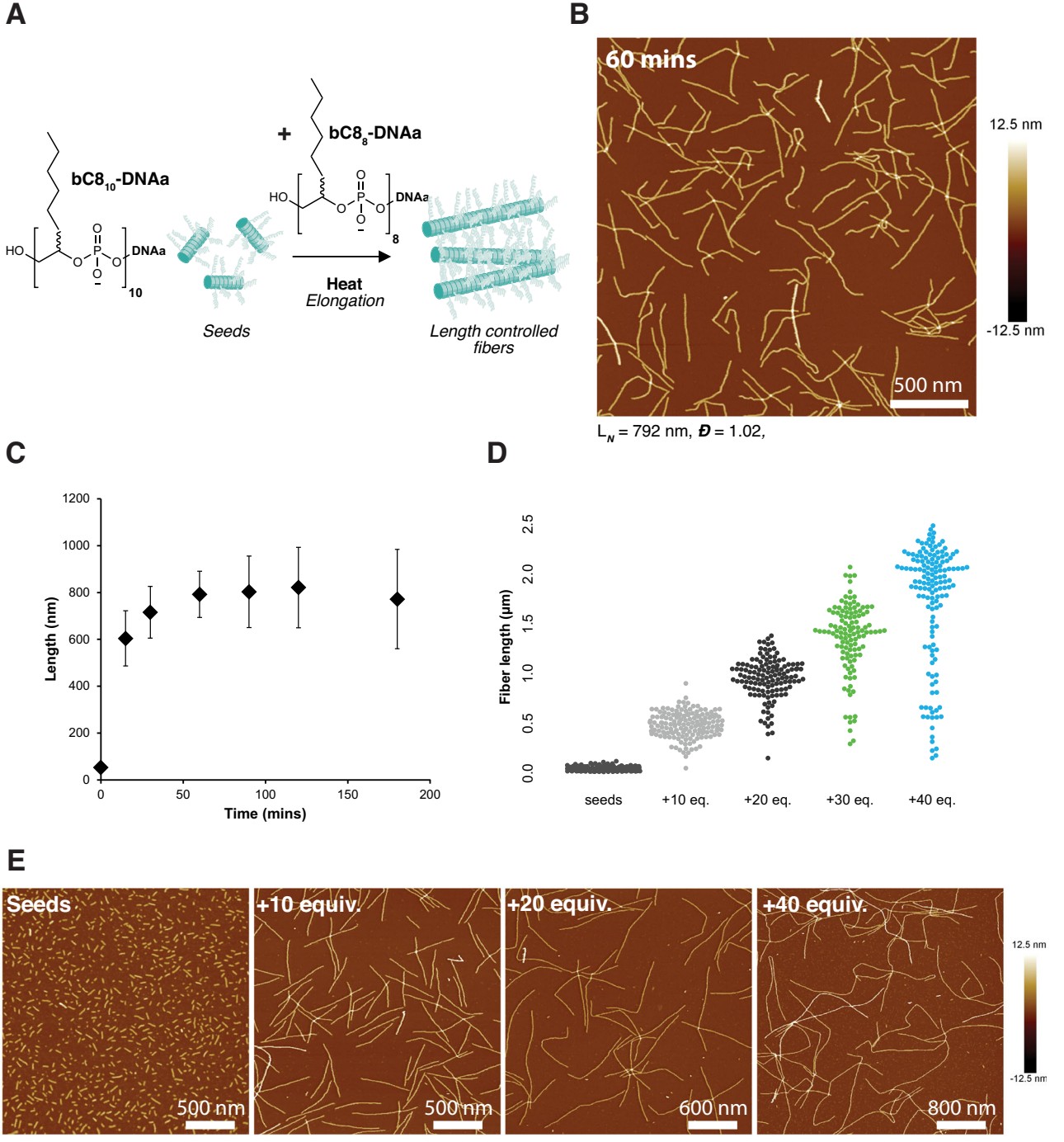

**Fig. 4 | Length control of DNA fibers using a seeded-growth mechanism. A** Pre-assembled short cylinders of **b**C8$_{10}$-DNAa serve as nuclei for the heat-driven elongation of bC8$_8$-DNAb fibers. **B** Representative image (AFM in the air on mica) of length-defined bC8$_8$-DNAa fibers. **C** Length of bC8$_8$-DNAb fibers following heating at 70 °C for varying times, measured by AFM. **D** Length of bC8$_8$-DNAb fibers is linearly dependent on the molar equivalents of bC8$_8$-DNAb to bC8$_{10}$-DNAa seeds.

Seeds: $L_N$ = 40 nm, Đ = 1.20, $N$ = 212; +10 equiv.: $L_N$ = 490 nm, Đ = 1.07, $N$ = 164; +20 equiv.: $L_N$ = 950 nm, Đ = 1.05, $N$ = 130; +30 equiv.: $L_N$ = 1360 nm, Đ = 1.07, $N$ = 117; +40 equiv.: $L_N$ = 1730 nm, Đ = 1.12, $N$ = 144. Linear fit: $L_N$ (nm) = 43.1$x$ + 44.7. RMSE = 30 nm. **E** Representative images (AFM in air on mica) of length-defined bC8$_8$-DNAb fibers with different equivalents of bC8$_8$-DNAb.

systems (Fig. 6D)[58]. The melamine-thymine interactions could potentially increase the rigidity of the DNA adjacent to the core in addition to non-covalently cross-linking the DNA oligomers, stabilizing the fiber assemblies[37,59,60].

## Discussion

Experimental and computational results suggest that the heat-activated and stabilized assembly of our amphiphilic DNA

oligomers is conditional on the hydrophobic segment curling into a helical shape, activating monomers for fiber formation. The stability of this curled conformation and its ability to pack into elongated supramolecular fibers is highly dependent on the balance between the length of the oligomer backbone and the length of the branched alkyl chain of each monomer. Following assembly, the fibers slowly re-equilibrated into short cylinders and spheres as the hydrophobic tail collapsed into a more globular state. By seeding the self-

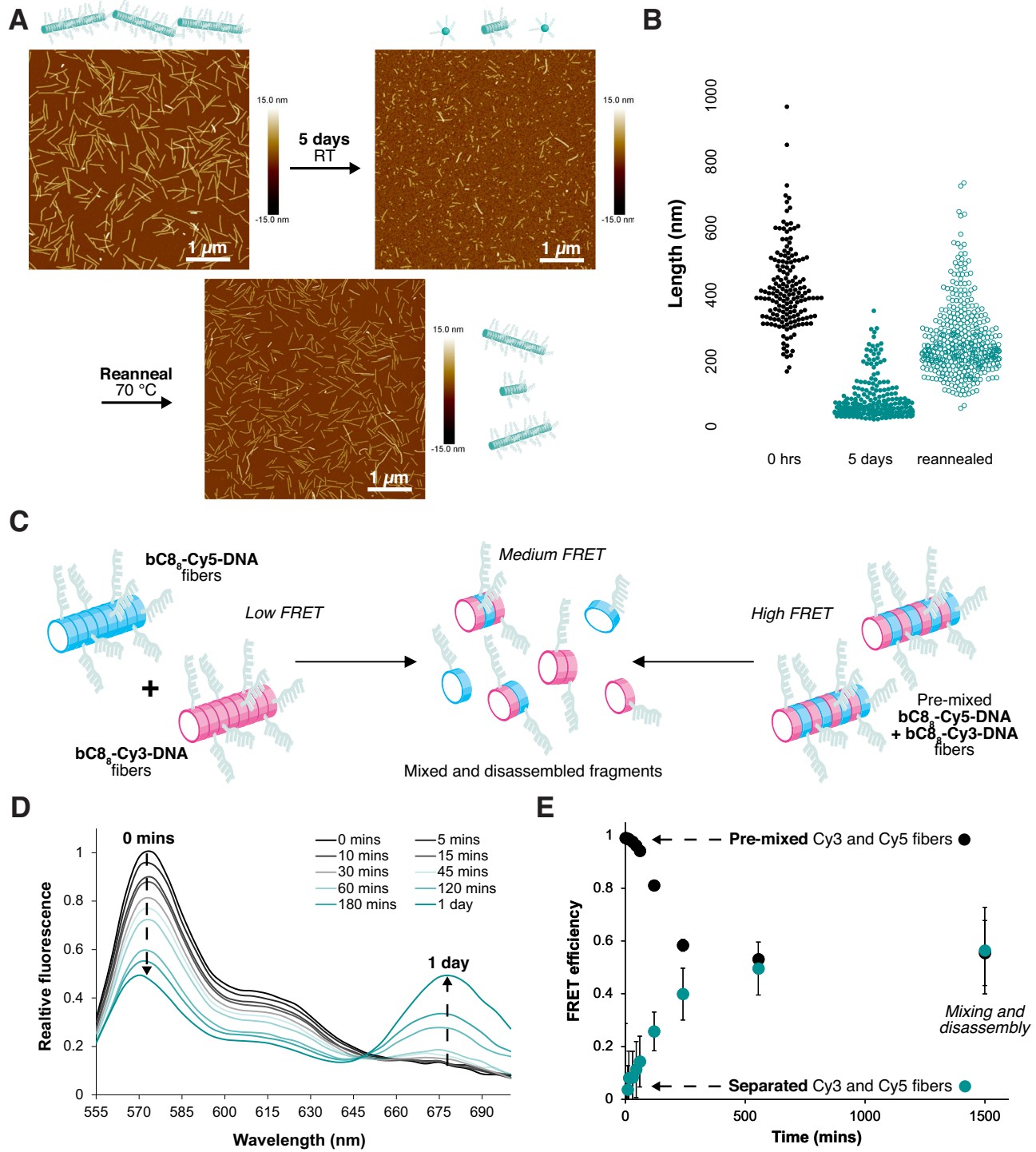

**Fig. 5 | Dynamics of DNA fibers. A** AFM in air on mica of bC8₈-DNAb fibers nucleated with bC8₁₀-DNAa 0 h following assembly ($L_N = 413$ nm, $Đ = 1.09$, $N = 169$), after 5 days at room temperature ($L_N = 74$ nm, $Đ = 1.6$, $N = 276$), and following reannealing ($L_N = 256$ nm, $Đ = 1.17$, $N = 414$). **B** Plot of fiber lengths of samples in (**A**). **C** Schematic of the FRET experiment performed to investigate dynamics. **D** Relative fluorescence emission spectra over time of bC8₈-Cy3-DNAa and bC8₈-Cy5-DNAa fibers that were assembled separately and then mixed. The increase in emission at 680 nm indicates FRET due to dye proximity. **E** FRET efficiency over time of separately assembled then mixed (green circles) and pre-mixed (black circles) Cy3 and Cy5-labeled fibers. Error bars represent standard deviation.

assembly with particles of a structurally related DNA oligomer, length-defined fibers with very low dispersity were achieved. The addition of a small molecule stabilized these metastable nanofibers at room temperature by forming hydrogen bond cross-links within their single-stranded DNA coronas, enabling their potential application in biomaterials and nucleic-acid therapeutics technologies.

Typically, block copolymer self-assembly theory has been based on discrete blocks with different chemical properties within a polymer; however, the presence of punctuating negative charges throughout the polymer backbone—as is the case with poly(phosphodiesters)—changes this paradigm and requires an updated theory that accounts for their high aqueous solubility and sensitivity to ions[61,62]. As demonstrated here, the observed behavior for these systems can be

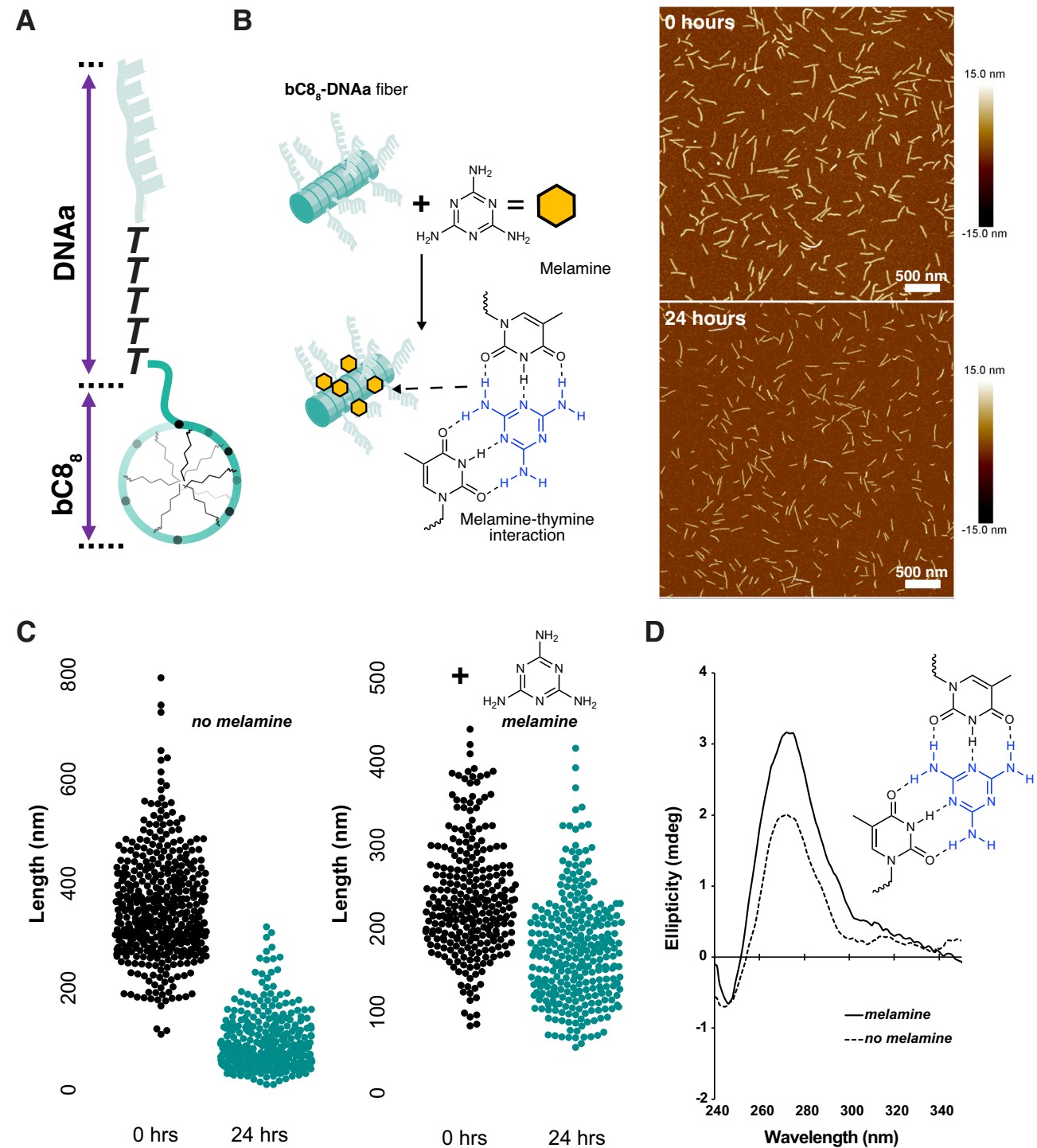

**Fig. 6 | Fiber stabilization using a small molecule. A** Detail of bC8$_8$-DNAa showing the location of the five thymine nucleotides. **B** Mixing assembled nucleated fibers with melamine inhibits their disassembly. AFM in air on mica of bC8$_8$-DNAa fibers nucleated with bC8$_{10}$-DNAa, immediately following melamine addition ($L_N$ = 230 nm, Đ = 1.08, $N$ = 276) and after 24 h ($L_N$ = 169 nm, Đ = 1.14, $N$ = 303). **C** Box plots of fiber lengths immediately following assembly ($L_N$ = 343 nm, Đ = 1.08, $N$ = 482) and after 24 h ($L_N$ = 85 nm, Đ = 1.37, $N$ = 389) for fibers without melamine, compared to those with melamine (data shown in **A**). **D** Circular dichroism of bC8$_8$-DNAa in the presence and absence of melamine.

the opposite of what would typically be expected, instead resembling the dynamic folding pathways seen in proteins. Further fundamental study of the self-assembly of amphiphilic poly(phosphodiesters), making use of MD simulations and systematic experimental designs, could help to better understand their mechanism of assembly and eventually allow for a robust prediction of morphology based on monomer sequence.

## Methods

### General assembly conditions

Samples were prepared in TAMg buffer. 1 X TAMg buffer consisted of 12.5 mM (CH$_3$COO)$_2$Mg.4H$_2$O and 40 mM Tris, with the pH adjusted to 8.0 using glacial acetic acid. 7.5 µM of DNA oligomer in 0.5 X TAMg was generally used for assembly unless otherwise stated in the text or in the methods below. Samples were heated from 25 to 99 °C at 0.5 °C per

minute, then removed from the heat and cooled to room temperature unless stated otherwise.

## Atomic force microscopy

About 5 µL aliquots were taken from the sample at the temperatures specified in the text and immediately deposited on freshly cleaved mica and left for ~15 s. The sample was then washed by adding 50 µL of filtered ultrapure water to the surface, which was immediately removed by touching the edge of the mica to filter paper. Washing was repeated up to four more times before a strong flow of compressed air was used to remove excess liquid from the sample for ~20 s. Samples were left under vacuum for at least 30 min prior to imaging. Imaging was performed under ambient conditions with a scan rate between 0.9 and 1.4 Hz.

## Total internal reflection fluorescence microscopy (TIRF)

About 9 µM of bC8$_8$-DNAa and 1 µM of bC8$_8$-Cy3-DNAa were mixed in 0.5 X TAMg (6.25 mM Mg$^{2+}$) and then heated at a rate of 1 °C per minute to 75 °C and held for 6 h before cooling to room temperature.

Glass coverslips were soaked in piranha solution (30% H$_2$O$_2$ solution and concentrated H$_2$SO$_4$, in a 1:3 v/v ratio) and sonicated for one hour. Coverslips were then etched by sonication for 30 min in a 0.5 M aqueous solution of NaOH (temperature maintained between 20 and 30 °C). Etched coverslips were rinsed three times with water (Hyclone molecular biology grade) and then sonicated in water for fifteen minutes. This procedure was then repeated with acetone (HPLC grade) to remove any surface-bound water molecules. Surface passivation with poly(ethylene glycol) silane was then carried out one coverslip at a time. The coverslip was dried under a nitrogen stream then a silicon mold was placed over the coverslip. A 99:1 (w/w) mixture of poly(ethylene glycol) silane (MW 5000) and biotin-poly(ethylene glycol) silane (MW 5000) was then dissolved in dry DMSO (25% PEG-silane w/w). The coverslip was preheated in an oven at 90 °C for 5 min then the freshly prepared PEG-silane/DMSO solution was added. After 15 min in the oven, the silicon mold was removed, and the coverslip was washed thoroughly with Hyclone water and dried under a stream of nitrogen.

Imaging chambers (~10 µL) were formed by pressing a custom polycarbonate film with an adhesive gasket onto the PEG-silane-coated coverslips. Two silicone connectors were glued onto the pre-drilled holes of the film to serve as inlet and outlet ports. Prior to imaging, 10 µL of a 1 mg/mL streptavidin solution was added to the surface and left to incubate for 10 min. The excess streptavidin was then washed with 50 µL 1 x TAMg buffer in Hyclone water prior to adding the sample to the surface.

Fluorescence imaging was carried out on a total internal reflection fluorescence microscopy (TIRF-M) setup consisting of an inverted microscope (Nikon Eclipse Ti) equipped with a motorized illuminator unit and Ti-ND6 Perfect Focus System (PFS). Samples were excited with the evanescent wave of either a 561 nm diode laser output or a monolithic laser combiner (Agilent Technologies, MLC-400B). The beam position was adjusted to attain total internal reflection through an oil-immersion objective (Nikon CFI SR Apochromat TIRF 100x, NA = 1.49). In all cases, the laser beam was passed through a multiband clean-up filter (ZET405/488/561/647x, Chroma Technology) and coupled into the microscope objective using a multiband beam splitter (ZT405/488/561/647rpc, Chroma Technology). Fluorescence light was spectrally filtered with an emission filter (ZET405/488/561/647 m, Chroma Technology). For imaging of Cy3, an additional emission filter (ET600/50 m, Chroma Technology) was used. 561 nm imaging was conducted at 1.0 mW, as measured from the objective.

The microscope and camera were controlled using the software NIS element from Nikon, capturing 16-bit 512 × 512 pixel images with an exposure time of 50 ms. All movies were recorded onto a 512 ×512 pixel region of a back-illuminated electron-multiplying charge-coupled

device (EMCCD) camera (iXon X3 DU-897-CS0-#BV, Andor Technology).

Samples were immobilized on the coverslip surface using a bio-tinylated DNA strand complementary to the fibers (sequence: 5' biotin-TATATGGTCAACTGAAAAA-Cy5 3'). This strand was added to the coverslip surface at a concentration of 10 nM to create a dense layer of biotinylated strands, which were then photobleached by irradiating at 30 mW using the 647 nm laser. About 10 µL of 5 nM fibers were then deposited onto the surface and washed with several 10 uL aliquots of 1 x TAMg prior to imaging. Images were converted to 8-bit RGB and colorized using ImageJ.

## In situ variable temperature fluorescence measurements

Fluorescence was measured using a RT-qPCR machine with a 96-well plate. About 25 µL of the sample (with 10% Cy3 labeled DNA oligomer) was used per well, and each condition was performed in triplicate. The temperature was increased from 25 to 99 °C at 1 °C per minute, while fluorescence of Cy3 was measured once per minute using the HEX channel.

## Cryo-EM sample preparation, data collection, and image processing

About 3 µL samples of bC8$_8$-DNAa and bC8$_{10}$-DNAa fibers, assembled in 0.5xTAMg buffer at a concentration of 10 mM, were applied to negative glow-discharged holey carbon grids (CF-2/1- 3CU) immediately after treating the grids with negative glow discharge in air for 15 mA for 30 s. Grids were flash-frozen in liquid ethane after blotting the excess solution once for 5 s with a blot force +3, with no wait or drain time, using an FEI Vitrobot Mark IV set at 4 °C and 100% humidity. A total of 1332 movies for the bC8$_8$-DNAa fiber and 2736 movies for the bC8$_{10}$-DNAa fiber were acquired using SerialEM software on a Titan Krios electron microscope operated at 300 kV. The samples were kept at −190 °C during imaging. Movies were collected in counting mode with 40 frames each in a Gatan K3 direct electron detector equipped with a Bioquantum imaging filter. Movies were collected using a total exposure of 60 e⁻/Å2 at a nominal magnification of 81,000 × corresponding to a calibrated pixel size of 1.09 Å and a defocus range from −0.84 to −2.24 µm.

All movies were processed using CryoSPARC v4.4.0[63]. Specifically, movies were first corrected using Patch Motion Correction. Contrast transfer function parameters were estimated using Patch CTF Estimation, and subsequently, Filament Tracer was used to pick an initial set of particles (103,204 particles for bC8$_8$-DNAa fiber, 179,584 particles for bC8$_{10}$-DNAa fiber). Low-quality particles were discarded using multiple rounds of 2D Classification, resulting in a final set of clean 2D classes (25,292 particles for bC8$_8$-DNAa fiber, 28,302 particles for bC8$_{10}$-DNAa fiber).

## Molecular dynamics simulations

Simulations were prepared using the System Builder from Schrödinger *Inc* and run with the *Desmond* MD engine[64]. All systems used the OPLS3 force field and SPC water molecules and were equilibrated in the NVT ensemble[65,66]. Production runs were performed in the NPT ensemble at either 300 or 358 K (1 bar/Martyna−Tuckerman−Klein coupling scheme) with a 2 fs timestep. Electrostatic interactions were calculated using the particle mesh Ewald method. Van der Waal and short-range electrostatic interactions were cut off at 9.0 Å. Production simulations were run for 300 ns. All simulations were run with GPUs on the DeepGreen supercomputer. Data analysis and visualization were performed with Matplotlib and PyMOL, respectively[67,68].

## Seeded growth of length-defined fibers

Seeds were generated prior to performing the seeded growth: 5 µM of bC8$_{10}$-DNAa in 0.25 X TAMg was heated at 95 °C for 5 min before being

slowly cooled to 4 °C over the course of 1 h. Seeds were stored at 4 °C prior to use. To generate length-defined fibers, previously assembled seeds were added to 7.5 μM of unassembled bC8$_8$-DNAb in 0.5 X TAMg to give molar ratios of 10, 20, 30, 40 and 50 bC8$_8$ to bC8$_{10}$. These mixtures were then heated at 70 °C for 60 min before being cooled to room temperature and analysed by AFM. Fiber lengths were measured manually using ImageJ software. Segmented lines were used to trace the fiber paths by hand. Every assembly within an AFM image was measured to remove measurement bias.

### Förster resonance energy transfer (FRET) studies

Samples were prepared in 0.5 X TAMg. Four separate fiber assemblies were first generated: "Cy3", "Cy5", "mixed", and "unlabeled". About 10 μM bC8$_8$-DNAb was combined with 1 μM bC8$_8$-Cy3-DNAa for "Cy3" fibers or 1 μM bC8$_8$-Cy5-DNAa for "Cy5" fibers. About 10 μM bC8$_8$-DNAb was combined with 0.5 μM bC8$_8$-Cy3-DNAa and 0.5 μM bC8$_8$-Cy5-DNAa for "mixed" fibers. "Unlabeled" fibers contained no dye-labeled DNA oligomers, only 10 μM bC8$_8$-DNAb. Pre-assembled bC8$_{10}$-DNAa seeds were then added to the unassembled "Cy3", "Cy5", "mixed", and "unlabeled" fibers to give a final molar ratio of bC8$_8$ to bC8$_{10}$ of 10 to 1 in each sample, before being heated at 70 °C for 1 h and cooling to room temperature to assemble.

Immediately following assembly, three experiments were set up directly in a 96-well plate. For the "separately assembled" experiment, 10 μL of "Cy3" fibers and 10 μL of "Cy5" fibers were mixed. For the "mixed" experiment, 20 μL of "mixed" fibers were added to the plate directly. Finally, a "no FRET" control was performed in which 10 μL of "Cy3" fibers was mixed with 10 μL of "unlabeled" fibers. Each experiment was performed in triplicate, and fluorescence was measured at 0, 5, 10, 15, 30, 45, 60, 120, 180 min, and 1 day following mixing. An excitation wavelength of 535 nm was used, and fluorescence intensity at 575 nm was measured.

### Dynamics−melamine stabilization

Seeded growth was carried out in 0.5 X TAMg using 10 μM of bC8$_8$-DNAa and 1 μM of pre-assembled bC8$_{10}$-DNAa seeds by heating at 70 °C for 1 h and cooling to room temperature. Melamine addition (20 mM in 1 X TAMg stock) was carried out either before the self-assembly heating stage or immediately after cooling the self-assembly solution to room temperature. In either case, the final melamine concentration in the solution was 10 mM. The control experiment was carried out in 0.5 X TAMg with just the pre-assembled bC8$_{10}$-DNAa seeds and melamine (20 M in 1 X TAMg added prior to the heating stage for a final solution concentration of 10 mM in 0.5 X TAMg). Aliquots for deposition onto freshly cleaved mica for AFM sample preparation and CD measurements were taken immediately after cooling (or post-cooling addition of melamine) as well as after aging the self-assembly solution for one day.

CD measurements were performed using a 1 mm path-length quartz cuvette on a Jasco-810 spectropolarimeter equipped with a xenon lamp, a Peltier temperature control unit, and a water recirculator. The data were collected using a scan rate of 100 nm min$^{-1}$ and a bandwidth of 1 nm with three accumulations per scan. A solution of 0.5 x TAMg was used as a blank.

## Data availability

Experimental procedures and supplementary items are provided in the Supplementary Information. All relevant data and additional details are available from the authors.

## Code availability

All code and input files needed to reproduce the simulations in this work can be accessed at https://github.com/zormanmarlo/DNA_amph_natcomm[69].

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

## Acknowledgements

This work was supported by the Natural Sciences and Engineering Research Council of Canada (NSERC), the Fonds de Recherche Nature et Technologies (FRQNT), and the Canada Foundation for Innovation (CFI). LC-MS and analysis by Dr. Alexander Wahba. Phosphoramidite synthesis was performed by Violeta Toader. Jianing Li and Marlo Zorman were supported by the NSF CAREER award (CHE-1945394 to J.L.).

Computations were performed on the Vermont Advanced Computing Core, supported in part by NSF award No. OAC-1827314. M.G.R. thanks the NSERC for a Vanier Scholarship. F.J.R. thanks the Australian Research Council (ARC) for a Discovery Early Career Research Award (DECRA). We thank Dr. Kaustuv Basu from the McGill Facility for Electron Microscopy (FEMR) for assistance during cryo-EM data collection. Cryo-EM work was supported by the Natural Sciences and Engineering Research Council (RGPIN-2017-05828) to A.G. A.G. holds the Canada Research Chair in Macromolecular Machines in DNA repair (Tier 1). Tianxiao Peter Yang is a recipient of a doctoral fellowship from the Fonds de Recherche Santé Québec (FRQS).

## Author contributions

M.D.D. and M.G.R. performed synthesis and purification of DNA oligomers, self-assembly of bC8$_8$ DNA fibers, seeded-growth experiments, and dynamic fiber studies using FRET. M.D.D., M.G.R., F.J.R., and H.F.S. contributed to the experimental design and wrote the paper. M.D.D. performed AFM of bC8$_{10,12}$ and bC12 DNA fibers, and variable temperature AFM and fluorescence studies. T.T. and P.X. carried out early self-assembly and characterization (AFM, DLS) of DNA fibers. M.G.R. performed studies on the melamine stabilization of fibers. C.M.P. and G.C. designed and performed the TIRF experiments. T.P.Y. and A.G. designed and performed the cryo-EM studies. M.Z. and J.L. designed and carried out the molecular dynamics simulations.

## Competing interests

The authors declare no competing interests.
