## [Peer Review File · Nature Communications]

Beyond Traditional Polymer Assembly: Heat-Activated Growth of Metastable, Length-Defined DNA FibersReviewers' Comments:

Reviewer #1:

Remarks to the Author:

The group observes the assembly of a type of DNA-copolymers into tubules, short cylinders and spheres. The objects termed "curlamers" by the authors react to heating and cooling. Remarkably, certain combinations of poly(phosphodiester) length and DNA nucleotides yield wildly varying assembly behavior. The experiments are sound and the manuscript is well written.

Overall I am rather divided about the novelty of this work. On one side, it is intriguing to dictate assembly of this special class of block copolymers into different shapes and even control the length distribution of assembled cylinders. Further, a model is provided to explain the behavior partly. One additional advantage of this material could be the possibility to make use of DNA sequences for binding of additional molecules. Unfortunately, authors do not make use of this DNA addressability. Instead they use melamine to form hydrogen bonds with thymine nucleotides to transiently stabilize the assemblies. Authors do not comment on why not complementary DNA was used here. Finally, references 35 and 36 of this manuscript reference two articles by author's group that describe similar types of molecules that also assemble into spheres and tubules, there also into sheets. I am aware that there is an additional level of control over the cylinder size here. However, I am not sure if this has the breakthrough character that Nature Communication is striving for. It is probably up to the editor to decide this.

Minor comments:

The choice of melamine over a complementary DNA strand to stabilize the DNA tubules should be explained.

I do not understand the sentence starting line 69: "Previously, we demonstrated ..." Delete "that"?

Reviewer #2:

Remarks to the Author:

The manuscript by Dore and co-workers presents a systematic study exploring the formation of DNA nanofibers from DNA-lipid amphiphiles triggered using heat. The DNA sequence stays the same, but the hydrophobic block region is changed to alter the constructs' "curlamer" conformation. High temperatures are required to unfold the oligomer, upon cooling, nanofibers or sphere-like macrostructures form which are highly dependent on the quantity and chain-length of lipids in the curlamer. The polymer structures display interesting properties with varying lengths, curvatures and dynamic behaviour. Using pre-folded seeds, the authors were able to control the formation of DNA nanofibers by varying the seed-oligomer molar ratio. Similar to protein polymers, the DNA nanofibers are in dynamic equilibrium and unfold in a matter of hours. Interestingly, the rate of unfolding can be slowed down by adding a stabilizing agent which complexes to the DNA.

The story is well written and presented. This work provides some attractive insights into polymer formation and control. However, the degree of novelty is not significant, a number of similar publications exist in the literature, including from the Sleiman group. Overall, I feel this is a very interesting study, but lacking novelty and application for publication in Nature Communications. I also have a number of questions that should be addressed:

- Figure 2A. Interestingly all the different oligomers form nanofibers but with markedly different lengths. The authors attribute this to the disk-like "curlamer" shape which dictates the main argument of the story. However the differences could also be due to other factors, for example, small differences in the degree of purity, age of samples or subtle changes in sample preparation. Have the authors

ruled these factors out? The synthesis and AFM characterization should be repeated more than once to show these trends are reproducible.

- Each of the curlamers could have different critical temperatures which could impact the downstream formation steps. The authors should determine they are above these transitions for each construct.
- Figure 1. TEM states average length of ~0.9 micron (line 109), yet AFM and CLSM show much larger lengths. Can the authors explain this? The authors should compare side-by-side AFM vs CLSM vs TEM, lengths and persistence lengths to help gauge the bigger picture.
- Why do you need magnesium and not just heat? Does the MD simulations also use Mg? Also the amount of magnesium is varied, in some instances its 6.5 mM, for TEM it is higher, this could impact the outcome of the macrostructure.
- Please add CLSM images showing higher contrast, the background fluorescence should be seen. In addition, many fibers should be in view not just isolated ones. This is required to help understand the different populations.
- SI should include side-by-side simulations of all the different disks. Currently its only 2 types.
- Line 103 – heated not heating. What does holding for 6 hours mean, at 75 degrees? Don't you need to heat above this temperature to get the fibers to form (line 96)?
- Line 191 Fibers stopped growing after 60 mins. Yet in Figure S13 LN numbers are increasing up to 120 mins.
- The curlamer has to be heated above 75 degrees to unfold the hydrophobic block. But the seed growth goes against this and states fibers can form below the critical temperature. Can you explain this?
- The authors should show hybridization to assembled DNA fibers using CLSM. This will help prove the orientation and demonstrate the single strand sequence is accessible.
- Figure S4 needs a control – Cy3 might be degrading at 90 degrees over extended periods of time.
- Please describe in more detail how the AFM length measurements were collected manually.
- Figure S5 and S9 DLS graphs should be the averages of multiple independent repeats, not just technical repeats.
- SI AFM scale bars are missing or not defined properly.
- SI Samples were purified using HPLC, 20-100 uL was injected with an OD 0.5. This seems really low?
- How do you calculate the extinction coefficients for each construct?

Reviewer #3:

Remarks to the Author:

The manuscript from the Sleiman laboratory furthers their understanding and exploration of the assembly of diblock copolymer-like molecules designed with one 'block' being a short, single-stranded DNA segment and the other 'block' being a short phosphodiester backbone with alkyl chain side groups of various lengths. The primary new understanding in this manuscript over their recent publications is the ability to control the length and polydispersity of chains formed via heating of dissolved molecules as well as via seeding of desired assemblies. The results and ability to control chain formation, produce segmented chains (from recent papers), and control dispersity reminds me of the Crystallization Driven Self-Assembly that people use in the literature to control micelle-like nanoparticle formation with essentially living ends that can be added to with addition of new blocky molecules. Since the chains discussed herein require heating (and multivalent cations) to form, the mechanism underlying this assembly seems to be more aligned with something similar to an LCST (lower critical solution temperature) transition or behavior observed in cold-denaturing proteins. But, the living-like assembly and control of the length and dispersity is similar in effect.

I typically find novelty arguments in reviews to be lazy responses to submitted manuscripts. However, in this case, the submission to Nature Communications begs the question-beyond the demonstration of more control of length/dispersity of chain formation, what is the novelty within the work beyond the 2022 JACS communication and 2021 Chem paper? More importantly, however, is the work needed to more definitively understand the nanostructure underlying the chains that are clearly formed with

Mg²⁺ and higher temperatures. This, I will dive into in more detail below:

In general, the AFM data in the manuscript seems to undermine the model of assembly proposed of the curling of individual phosphoesters into disks that then stack to form the core of a cylinder that they have a dilute corona of DNA extending from the highly negatively charged phosphodiester backbone. In looking at the diameters and heights of the chains in AFM, the following becomes apparent. In figure 2, bC8-10 and bC8-12 seem to have similar diameters from AFM. This is expected since the change in disk diameter by adding two more backbone units would be quite small. However, bC12-10 and bC12-12 appear to have quite different cross-sections with the shorter molecule bC12-10 having much larger diameter. Even assuming typical AFM tip convolution effects, the differences in diameters between these two molecules' chains is clear, implying a different cross-section or different assembly mechanism since bC12-12 should have the larger cross-section than bC12-10.

Also in Figure 2, the bC8 AFM data suggests something different than what is described in the manuscript. The bC8-12 "spheres" in 2A appear to be very short segments of some kind of similar chain assembly as seen in the bC8-10 and -8, just very short segments of the chains. At the very least, many of the observed nanoparticles are anisotropic in morphology. Similarly, the "spheres" in bC12-8 molecules, while much smaller in cross-section than the chain assemblies and obviously a different morphology, appear to regularly have cavities/holes within the nanoparticles. Perhaps this is due to drying the particles for imaging. However, no similar cavities appear in the chains.

The TEM in 1C is not too helpful in telling the reader anything beyond what the AFM data provides. The drying of the thin films of fibrillar structures displays a hierarchy of diameters/lengths while the AFM reveals individual chains that can be analyzed. If one were to include electron microscopy that could help elucidate what the local nanostructure is, then an in situ imaging technique like cryoTEM would be an excellent complement to the AFM data.

Even with the AFM data, more analysis could be performed in order to better understand the local structure of the 1-D, fibrillar assemblies. For example, some basic line tracing of the nanostructure could provide a more quantitative comparison of the nanostructure and, particularly through the heights measured, could shed light on a different assembly mechanism. For example, it seems through heights measured and diameters that the cross-section of the fibrils is much larger than what would be expected for the proposed disk assembly. This may lead to proposal of a thermodynamically similar assembly mechanisms (e.g., LCST-like collapse of the hydrophobic alkyl chains causing aggregation) but with a much different core (e.g. extended phosphodiester blocks or chain folded blocks in the core domain). These type of mechanisms/model would allow for more molecules to reside together along the length of the chain providing both a larger cross-section as well as more DNA chains extending into solution as corona chains. The lengthscales/sizes of fibrillar cross-section with the proposed disk core are much too small relative to the observed data. Data from previous papers (e.g., the > 3 nm diffraction-like peak in x-ray scattering) supports the point that there is some longer lengthscale regular packing of longer molecules in the cores and not a single molecule hydrophobic disk that is both very small in diameter as well as in length along the cylinder fibril. The melamine stabilization of fibers would presumably provide the stability needed so that in-depth study of nanostructure/cross-section could take place so that fibril morphology is thoroughly understood. Additional support for a different packing mechanism in the core of the nanostructure is the fact that the authors observe the following when comparing bC8 and bC12. As stated, "This trend from fibers to spheres as the number of bC12 monomers 148 decreased from 12 to 8 is the opposite to that observed for bC8, where spheres are formed with 12 149 monomers and the fibers with 8. Evidently, the hydrophobic to hydrophilic ratio alone is not 150 governing the assembly morphology." Rather than simply state the fact that this puzzle exists, it would be more illuminating for the reader for the authors to propose a packing model that helps describe the actual data and nanostructures and their relative sizes/geometries and not to fall back on the previously proposed cylinder core due to the "curlamer" assembly.

Another aspect of the assembly requires additional thought and will also help with the comments made above about the structure underlying the fibrillar assembly. In the 2021 Chem paper (reference #40), it was shown that bC12-12 shows no assembly without Mg²⁺ (or Ca²⁺) present. Obviously, Mg²⁺ is an important aspect of the assembly in the current manuscript. Unlike the discussion in the other manuscript that only described possible "screening" of electrostatic charges in the phosphodiester block backbone, it is likely the multivalent cations are acting as some type of physical crosslinking agent between the multiple negative charges and acting as an active part of the nanostructure formation in the fibril formation. This would be more favorable at low temperature, but, the fact that assembly into fibrillar structure does not occur both at low temperatures as well as with a lack of divalent cation strongly suggests these ions act as ionic crosslinkers to the phosphodiester backbone charged groups. A simple experiment would be to perform assembly with monovalent salts at similar concentrations and/or ionic strengths and then observe if fibrillar assembly occurs. If it does, then charge screening is primarily the roll of the ions. If not, then there is direct evidence of the importance of the divalent ions in the mechanism of assembly. My hypothesis is that monovalent salts provide a completely different assembly route (if one exists at all). For reference, there are many papers in the literature that describe the complexation of polyelectrolytes with oppositely charged, multivalent ions to form coacervates, crystals, micelles, hydrogel network, collapsed brushes at interfaces, amongst others.

As described earlier, the control of length of fibers with seeding/temperature is a bit incremental when the assembly process has been thoroughly discussed in manuscripts over the past year or more. Perhaps if combined with more nanostructure characterization (e.g. higher rez/mag AFM with line traces, sharper tips to quantify diameter and height, TEM (or cryoTEM) revealing in situ diameters, SAXS of dilute samples using cylinder or cylinder micelle models vs other geometry of cross-nanostructure cross-section for fitting of fibrillar diameter) and a refinement of the model for how molecules are packing within the fibrillar nanostructure, there would be a solid foundation for publication. The simulations performed on individual molecules that show collapse of the hydrophobic side chains at higher simulation temperature is in support of the temperature-induced, LCST-like assembly. However, the purposeful packing of the molecules together into a cylinder core of individual disks is but one version of nanostructure that sequesters the alkyl chains in the core. More straightforwardly, the simulated nanostructure is much smaller in cross-section than the chains observed via AFM. Prior to publication, more thought and analysis should be brought to bear on the nanostructure in order to proposed a more physically meaningful and accurate nanostructure that is supported by the experimental data.

REVIEWER COMMENTS

Reviewer comments are shown in ***bold italic***.

Reviewer #1 (Remarks to the Author):

The group observes the assembly of a type of DNA-copolymers into tubules, short cylinders and spheres. The objects termed "curlamers" by the authors react to heating and cooling. Remarkably, certain combinations of poly(phosphodiester) length and DNA nucleotides yield wildly varying assembly behavior. The experiments are sound and the manuscript is well written.

Overall I am rather divided about the novelty of this work. On one side, it is intriguing to dictate assembly of this special class of block copolymers into different shapes and even control the length distribution of assembled cylinders. Further, a model is provided to explain the behavior partly. One additional advantage of this material could be the possibility to make use of DNA sequences for binding of additional molecules. Unfortunately, authors do not make use of this DNA addressability. Instead they use melamine to form hydrogen bonds with thymine nucleotides to transiently stabilize the assemblies. Authors do not comment on why not complement DNA was used here. Finally, references 35 and 36 of this manuscript reference two articles by author's group that describe similar types of molecules that also assemble into spheres and tubules, there also into sheets. I am aware that that there is an additional level of control over the cylinder size here. However, I am not sure if this has the breakthrough character that Nature Communication is striving for. It is probably up to the editor to decide this.

We thank the reviewer for their supportive comments regarding the science and its presentation in this manuscript. However, we also understand that the novel parts of this work were not presented adequately in the original submission. To address this, we have made significant changes to the manuscript to highlight and support the novelty of the work.

First, we have further emphasized the discovery of the unique heat-activated, seeded-growth of the DNA-oligomers. This has been done by putting its novelty into context by considerably editing the introduction and conclusion, and importantly, by performing additional experiments. Variable temperature fluorescence measurements using a real-time PCR instrument was carried out on **bc8₈-**, **bc8₁₀-**, and **bc8₁₂-DNAa**, and new AFM images were also collected. These experiments afforded additional insights into the assembly mechanism of the DNA-oligomers:

- Heating rate rather than cooling rate controlled the morphology. Slower heating rate resulted in longer one-dimensional structures for all three oligomers compared to a fast-heating rate, but the same trend between monomers was observed: **bc8₈-DNAa** gave the longest fibers and **bc8₁₂-DNAa** the shortest (**Figures 2A, S2, S3, S10, S12, S18**). This supports the hypothesis of a heat-activated nucleation-growth mechanism, where rates for nucleation and elongation increase with temperature.
- All three oligomers showed an assembly transition as temperature increased (**Figure 1C, S11**) with **bc8₈-DNAa** having the highest transition and **bc8₁₂-DNAa** the lowest. The lower transition temperatures of the more hydrophobic oligomers suggested a higher driving force for assembly; however, they also showed a lower propensity to form long fibers by AFM (**Figure 2A**). This is due to the less ideal geometry of the curled oligomeric segments for a fiber morphology compared to **bc8₈-DNAa**, as supported by cryogenic electron microscopy described below (**Figure 2B, 2C, 3C**).
- **bc8₈-DNAa** also had the sharpest transition, indicating the highest degree of cooperativity. This indicates a more efficient fiber elongation process due to its more ideal packing geometry.

Heat-activated seeded growth is extremely unusual for supramolecular polymers, due to their typical dependence on intermolecular interactions such as hydrogen bonding that weaken at high temperatures. Here, the supramolecular fibers actively nucleate and elongate at high temperatures rather than low, reversing the normal paradigm. To the best of our knowledge, there are no other examples of heat-activated seeded growth of soft supramolecular fibers in the literature. Our DNA

system may find use in materials in an analogous way to polymers with a lower critical solution temperature, such as PNIPAM. Furthermore, the assembly can be seeded, allowing us to control the dimensions of the fibers and hence the properties of the material.

Second, we have carried out cryogenic electron microscopy (cryo-EM) on our fibers and gained much greater insight into the assembly mechanism:

- 2D particle analysis was performed, providing sub-nanometer scale information on the structure of the fibers. The data strongly supports our hypothesis that the DNA-oligomers assume a conformation in which the hydrophobic blocks curl into a disc-like geometry, that stack along the direction of the long fiber axis.
- Furthermore, a slight twist in the fiber core is evident in the cryo-EM data, suggesting that the disc-like geometry has a degree of helicity. Less order is observed in **bc8₁₀-DNAa** compared to **bc8₈-DNAa**, explaining its lower propensity to form long fibers.
- **bc8₁₀-DNAa** fibers also have a wider core diameter, congruent with our geometric model that the oligomer curls in a way that leads to a mismatch between core diameter and length of the **bc8** alkyl chain. These results are presented in **Figures 2B, 2C, 3C**.

Overall, this new data, coupled with the previous observations reveal that the self-assembly process deviates from the traditional principles of microphase separation. As the hydrophobic component is increased, instead of becoming more stable, the fiber morphology becomes destabilized. This unexpected behavior stems from the role that the specific sequence and length play into the stability of the oligomer “curled” conformation. Very few studies have examined the effect of polymer *sequence* on self-assembly: our study uncovers new rules that sequence-controlled polymers follow, due to the precision of their chain length and sequence.

The reviewer also suggested using the DNA block to bind other molecules to the fibers. This is an interesting idea and will certainly be a future direction. Here, we argue that the use of a small molecule to stabilize the assemblies is a novel and simple way to gain function from the DNA block by increasing its thermal stability. Melamine has previously been used as a small molecule chaperone to assist in the assembly of poly-thymine duplexes comprising *anti-parallel* stretches in DNA tiles (**Reference 58** in revised manuscript). Here, we show that this approach can be extended to stabilize supramolecular architectures wherein DNA base-pairing is *not the primary* supramolecular interaction driving self-assembly, and wherein the DNA blocks in adjacent amphiphiles are not predisposed to adopting anti-parallel configurations.

Figures with new data are presented below accompanied by brief summaries of the additional mechanistic insights gained from them:

Figure 3. Internal structure and dimensions of DNA fibers. **A.** Molecular dynamics simulations show that a single $bC8_8$ -DNAa oligomer assumes a more structured disc-like conformation at 85 °C compared to room temperature (RT) (images at 300 ns). Phosphates shown in red and orange, backbone in black, alkyl chains in grey. **B.** Structure of this class of DNA-oligomers with parameters L (the length of the monomer's alkyl chain) and N (the number of monomers) marked. **C.** Proposed curled oligomer conformation and fiber structure dimensions based on the cryo-EM analysis. Blue line with black dots represents the polyphosphodiester backbone of the oligomer. The slightly helical, curled conformation of the $bC8_8$ block is shown, as is the packed DNA block. Scale bar: 5 nm.

NEW MECHANISTIC INSIGHTS (Figure 3): Sub-nanometer scale resolution of the fiber structure showing the dimensions of the core and corona regions, evidence of a fold in the DNA portion, and a slight helicity in how the oligomers are packed.

Figure 1. Mechanism of supramolecular fiber assembly of **bC8₈-DNAa**. **A.** Molecular structure of **bC8₈-DNAa** and scheme illustrating its proposed self-assembly mechanism into supramolecular fibers as temperature is increased. **B.** AFM images (in air on mica) of **bC8₈-DNAa** at different temperatures during heating. **C.** Normalised fluorescence of **bC8₈-Cy3-DNAa** during incremental heating from 20 to 99 °C. **D.** TIRF-M image of **bC8₈-Cy3-DNAa** after slow heating to 75 °C and holding for 6 hours.

Figure S5: Normalized fluorescence and derivative of fluorescence (dashed line) of **bC8₈-Cy3-DNAa** with 6.25 mM Mg²⁺ or water as temperature was increased from 25 to 99 °C at 0.5 °C per minute. Detailed view of 80 to 99 °C shown in Fig. 1C.

Figure S11: Fluorescence vs temperature curves of DNA oligomers heated at 0.5 °C / min from 25 to 99 °C. 7.5 μM DNA-oligomer in 6.25 mM Mg²⁺. C8-8 corresponds to **bC8₈-Cy3-DNAa**, C8-10 to **bC8₁₀-Cy3-DNAa** and C8-12 to **bC8₁₂-Cy3-DNAa**. C8-8 data is reproduced from Figure 1 as comparison. Fluorescence measurements showed self-assembly transitions at 86 °C for **bC8₁₀-Cy3-DNAa** and 84 °C for **bC8₁₂-Cy3-DNAa**, compared to 92 °C for **bC8₈-Cy3-DNAa**. This trend suggested a higher propensity for assembly as the number of **bC8** monomers increased, despite the observed decrease in fiber length. The lower assembly temperature for **bC8₁₀-Cy3-DNAa** and **bC8₁₂-Cy3-DNAa** could be due to their higher hydrophobicity; however, the reduced lengths of the resultant fibers signal less efficient molecular packing as hydrophobicity increased.

NEW MECHANISTIC INSIGHTS (Figures 1, S5, S11): More direct and precise evidence of the heat-activated assembly of **bC8₈**-, **bC8₁₀**-, and **bC8₁₂-DNAa** using a real-time PCR machine to perform variable temperature fluorescence experiments. Different transition temperatures were identified for the three oligomers with **bC8₈-DNAa** having the highest and sharpest. This suggested a lower driving force

for assembly due to the lower hydrophobicity, but a more cooperative assembly mechanism due to the more efficient packing.

Figure 2. Fiber formation is dictated by oligomer conformation. **A.** bC₁₀-DNAa and bC₁₂-DNAa form short fibers and very short fibers respectively, in contrast to the long fibers formed from bC₈-DNAa. AFM images in air on mica. **B.** Cryo-EM visualization of the bC₈-DNAa and bC₁₀-DNAa fibers. Representative cryo-EM micrographs and 2D classes of the bC₈-DNAa fibers (**B**) and the bC₁₀-DNAa fibers (**C**). The major 2D classes were used to measure the diameter of the fibers and their core. The dimensions of the fibers are indicated, and white arrowheads mark the wisps protruding from the central core. Scale bars are 500 and 5 nm, respectively.

Figure S2: bC8₈-DNAa at temperatures below assembly. Samples were heated from 25 to 99 °C at 0.5 °C per minute. Aliquots were deposited on mica at indicated temperatures then immediately washed and dried. AFM in air on mica. 7.5 μM DNA-oligomer in 6.25 mM Mg²⁺.

Figure S3: bC88-DNAa at temperatures above assembly. Samples were heated from 25 to 99 °C at 0.5 °C per minute. Aliquots were deposited on mica at indicated temperatures then immediately washed and dried. AFM in air on mica. 7.5 μM DNA-oligomer in 6.25 mM Mg²⁺.

Figure S10: DNA oligomers heated at 0.5 °C per minute from 25 to 99 °C, then quickly cooled to room temperature. AFM in air on mica. 7.5 μM DNA-oligomer in 6.25 mM Mg²⁺.

Figure S12: DNA oligomers heated at 1 °C per minute from 25 to 99 °C, then cooled to room temperature at 1 °C per minute. AFM in air on mica. 7.5 μM DNA-oligomer in 6.25 mM Mg²⁺.

Figure S18: DNA oligomers at room temperature prior to any heating. AFM in air on mica. 7.5 μM DNA-oligomer in 6.25 mM Mg^{2+} .

NEW MECHANISTIC INSIGHTS (Figures 2, S2, S3, S10, S12, S18): Heating rate rather than cooling rate determines the morphologies of **bC8₈-**, **bC8₁₀-**, and **bC8₁₂-DNAa**. Slow heating leads to longer assemblies for all three oligomers, however the trend of **bC8₈-DNAa** forming the longest fibers and **bC8₁₂-DNAa** the shortest contradicts typical block-copolymer self-assembly theory. Less efficient packing was observed by cryo-EM for **bC8₁₀-** compared to **bC8₈-DNAa**, explaining this trend.

Minor comments:

The choice of melamine over a complementary DNA strand to stabilize the DNA tubules should be explained.

Complementary DNA would not be suitable for this purpose as it can only bind a complementary sequence with the correct directionality (antiparallel) and on a 1 to 1 basis per nucleotide. Melamine on the other hand can interact with nucleotides on a 1 to 2 basis, with parallel DNA directionality, and is not constrained by a bulky, charged poly(phosphodiester) backbone. The combination of the short thymine repeat region in **DNAa** and the 1 to 2 stoichiometry of melamine results in a network of hydrogen bonds within the fibers. This explanation was added to the manuscript on page 16.

*"We hypothesised that in this highly DNA-dense environment, melamine may form a network of intermolecular hydrogen bonding interactions between thymine in neighbouring **bC8₈-DNAa** oligomers, stabilizing the fibers and preventing disassembly. The use of melamine for this purpose is advantageous over a complementary DNA sequence due to its ability to interact with DNA through two faces at the same time. In combination with melamines small size and lack of bulky poly(phosphodiester) backbone, this gives the system greater degrees of freedom to form a hydrogen bonding network."*

We have also added more detail to **Figure 6A, 6B** to help illustrate this.

Figure 6. Fiber stabilization using a small molecule. **A.** Detail of **bC8₈-DNAa** showing location of the five thymine nucleotides. **B.** Mixing assembled nucleated fibers with melamine inhibits their disassembly. AFM in air on mica of **bC8₈-DNAa** fibers nucleated with **bC8₁₀-DNAa**, immediately following melamine addition ($L_N = 230$ nm, $D = 1.08$, $N = 276$) and after 24 hours ($L_N = 169$ nm, $D = 1.14$, $N = 303$). **C.** Box plots of fiber lengths immediately following

assembly ($L_N = 343$ nm, $D = 1.08$, $N = 482$) and after 24 hours ($L_N = 85$ nm, $D = 1.37$, $N = 389$) for fibers without melamine, compared to those with melamine (data shown in **A**). **D**. Circular dichroism of **bC88-DNAa** in the presence and absence of melamine.

I do not understand the sentence starting line 69: "Previously, we demonstrated ..." Delete "that"?

We have removed "that" from this sentence.

Reviewer #2 (Remarks to the Author):

The manuscript by Dore and co-workers present a systematic study exploring the formation of DNA nanofibers from DNA-lipid amphiphiles triggered using heat. The DNA sequence stays the same, but the hydrophobic block region is changed to alter the constructs' "curlamer" conformation. High temperatures are required to unfold the oligomer, upon cooling, nanofibers or sphere-like macrostructures form which are highly dependent on the quantity and chain-length of lipids in the curlamer. The polymer structures display interesting properties with varying lengths, curvatures and dynamic behaviour. Using pre-folded seeds, the authors were able to control the formation of DNA nanofibers by varying the seed-oligomer molar ratio. Similar to protein polymers, the DNA nanofibers are in dynamic equilibrium and unfold in a matter of hours. Interestingly, the rate of unfolding can be slowed down by adding a stabilizing agent which complexes to the DNA.

The story is well written and presented. This work provides some attractive insights into polymer formation and control. However, the degree of novelty is not significant, a number of similar publications exist in the literature, including from the Sleiman group. Overall, I feel this is a very interesting study, but lacking novelty and application for publication in Nature Communications. I also have a number of questions that should be addressed:

We thank the reviewer for appreciating the story and insights gained from our manuscript. Major changes have been made to the manuscript – including additional experiments – in order to strengthen the novelty. These include further investigation of the heat-activated nucleation-growth mechanism of the system using variable temperature fluorescence measurements and atomic force microscopy, and cryo-EM with 2D particle analysis to elucidate the structure of the fibers. These experiments and their findings are outlined in the response to Reviewer #1 above. The findings further strengthened our conclusions that the DNA-oligomer curls into a disc-like conformation with a slight helical nature when assembled into fibers at high temperatures.

We would like to clarify that one of the major findings of this manuscript is that the **assembly of the DNA-oligomers occurs upon heating**, rather than cooling (as the reviewer stated above). Upon cooling, the assemblies are metastable, but slowly disassemble over the course of 24 hours. This is unique as most supramolecular assemblies depend upon specific enthalpic interactions such as hydrogen bonds, which are weakened at high temperatures. Here, the high temperature results in a conformational change that causes fiber growth through the hydrophobic effect, resembling entropically driven assemblies seen in nature, such as collagen. Importantly as well, we find that the self-assembly does not obey the conventional rules of microphase separation. Upon increase of the hydrophobic component, the fiber morphology is de-stabilized, rather than stabilized; cryo-EM and modelling experiments support the new notion that the precise sequence and length of the hydrophobic component play a crucial role in the stability of the oligomer "curled" conformation. Very few studies have examined the effect of polymer *sequence* on self-assembly: our study uncovers new rules that sequence-controlled polymers follow, due to the precision of their chain length and sequence.

Figure 2A. Interestingly all the different oligomers form nanofibers but with markedly different lengths. The authors attribute this to the disk-like "curlamer" shape which dictates the main argument of the story. However the differences could also be due to other factors, for example, small differences in the degree of purity, age of samples or subtle changes in sample preparation. Have the authors ruled these factors out? The synthesis and AFM characterization should be repeated more than once to show these trends are reproducible.

The reviewer highlights an important point, that purity of the material is important for understanding self-assembly. As shown in the SI, the HPLC traces and mass spectrometry data indicate very high purities (> 95 %) (SI-II-d). All syntheses, assemblies, and AFM experiments were performed multiple times, and independently by different people. The data presented are a mix from these different experiments. Purity was always confirmed via HPLC and MS. A detailed height analysis has been performed from the AFM in air images and is presented in **Figures S24, S25**, and **Table S1, S2**. The dimensions of **bC8₈-DNAa** and **bC8₁₀-DNAa** were also studied using cryo-EM, and were consistent with the conclusions taken from AFM (**Figure 2B, C**). In summary, a slightly larger height was observed for **bC8₁₀-DNAa** by AFM and a slightly larger core diameter by cryo-EM. This supports our hypothesis that the core is slightly

wider due to the geometry of the curled disc, which is corroborated by other cryo-EM features (please see response to reviewer 1).

Figure S24: Representative AFM images used to measure height of nanostructures. Green lines indicate measurement sections. Nanostructures were assembled by heating slowly (1°C / min) to 99 °C and cooling quickly.

Figure S25: Comparison of height statistics from measurements detailed in Figure S22. For **bC8₁₂-DNAa**, two populations with different heights were observed: short fibers with a height of 5.66 (± 0.30) nm and very short, higher cylinders at 9.03 (± 0.53) nm. The greater diameter of the very short cylinders and their inability to elongate into fibers suggests a more dense, globular conformation of the oligomers. The lower height of the short fibers could indicate a structure in which the branched oligomeric core assumes a spiral conformation to maintain a smaller core diameter, resulting in a lower density of DNA in the corona and hence a lower height by AFM. The short length of these fibers suggests that this conformation is less conducive to forming long fibers.

Table S1: Summary of AFM characterization data for **bc8** DNA-amphiphiles. Assembly protocol: Heat 0.5 °C / min to 99 °C, cool quickly to room temperature (slow heat, fast cool). Measurements were taken along the fibers. Six measurements were taken for each morphology, where N = the total number of data points within those measurements).

Oligomer	Height (nm)	L/N	Morphology
C8₈-DNAa	6.41 (± 0.35), N = 372	1	Long fibers
C8₁₀-DNAa	6.75 (± 0.41), N = 773	0.8	Short fibers
C8₁₂-DNAa	high: 9.03 (± 0.53), N = 143 low: 5.66 (± 0.30), N = 372	0.67	Very short fibers (low) and shorter cylinders (high)

Table S2: Summary of DLS and AFM characterization data for various branched DNA-amphiphiles. Assembly protocol: Heat directly to 95 °C, cool 1 °C / min to room temperature (fast heat, slow cool). *D*: diffusion coefficient; % Pd: percentage polydispersity. DLS Measurements performed in triplicate. AFM measurements were taken as cross-sections.

Oligomer	D (cm ² /s) E-8	% Pd	Height (nm)	Diameter (nm)	L/N	Morphology
C12₁₀-DNAa	7.08 (± 1.68)	30.5 (± 11.6)	8.5 (± 0.6), N = 30	27.5 (± 1.8), N = 60	1.2	shorter fibers
C12₈-DNAa	29.7 (± 0.23)	13.9 (± 4.4)	collapsed	21.4 (± 2.2), N = 54	1.5	spheres
C8₈-DNAa	4.59 (± 0.30)	29.0 (± 14.3)	6.1 (± 0.3), N = 60	27.2 (± 1.9), N = 60	1	fibers
C8₁₀-DNAa	21.4 (± 0.11)	14.3 (± 7.7)	6.2 (± 1.1), N = 30	26.4 (± 1.4), N = 60	0.8	short cylinders
C8₁₂-DNAa	21.0 (± 0.02)	35.0 (± 3.8)	high: 9.5 (± 0.6), N = 30 low: 5.8 (± 1.1), N = 30	cylinders: 30.6 (± 3.0), N = 18 spheres: 24.1 (± 2.3), N = 60	0.67	short cylinders, spheres

Each of the curlamers could have different critical temperatures which could impact the downstream formation steps. The authors should determine they are above these transitions for each construct.

To answer this question, we performed new variable temperature fluorescence studies using a real-time PCR instrument. The transition temperatures for **bc8₈-DNAa**, **bc8₁₀-DNAa**, and **bc8₁₂-DNAa** were measured (**Figure 1C, S11**). Lower transition temperatures were observed for **bc8₁₀-DNAa** and **bc8₁₂-DNAa** relative to **bc8₈-DNAa**, indicating that the assembly protocols used in the original manuscript (primarily fast heating to 99 °C followed by slow cooling to room temperature) were in fact exceeding the transition temperatures for each assembly. In the revised manuscript, we have improved the general assembly heating protocol, informed by the new variable temperature fluorescence experiments. A slow heating from 25 to 99 °C has been used consistently for the **bc8** samples. This slow heating allows the assemblies to reach their thermodynamic minimum state (analogous to a slow cooling in typical systems). Longer fibers were observed for **bc8₁₀-DNAa** than previously observed when a fast heating and slow cooling protocol was used, but they were still much shorter than for **C8₈-DNAa** (**Figure 2A**). These experiments were added to the manuscript and SI (**Figure 1B, 2A, S2, S3, S10, S12, S18**).

Figure 1. TEM states average length of ~0.9 micron (line 109), yet AFM and CLSM show much larger lengths. Can the authors explain this? The authors should compare side-by-side AFM vs CLSM vs TEM, lengths and persistence lengths to help gauge the bigger picture.

The discussion surrounding lengths was confusing in the original submission. The ~ 0.9 micron length was in fact a reference to lengths measured by AFM following a specific heating protocol (**Figure S2** in original manuscript). That length had a very high dispersity, meaning much longer and much shorter

fibers were also present. The fluorescence microscopy (TIRF) images are from a heating protocol that favours very long fibers by heating very slowly. For clarity, the results presented in the manuscript have been changed to show simpler, more uniform heating protocols. The general protocol is now a slow heating followed by a fast cooling, unless otherwise stated. This is explicitly described in the manuscript on page 4, lines 98 to 125, and page 6: lines 137 to 155. Additional AFM images have been added to the SI to show the bigger picture.

Why do you need magnesium and not just heat? Does the MD simulations also use Mg? Also the amount of magnesium is varied, in some instances its 6.5 mM, for TEM it is higher, this could impact the outcome of the macrostructure.

Due to the high negative charge of the poly(phosphodiester) backbone along the entire lengths of the DNA-oligomers there is high intermolecular charge repulsion. Mg^{2+} ions screen the negative charges and allow self-assembly to occur. Divalent cations like Mg^{2+} are more efficient at charge screening than monovalent cations like Na^+ due to their higher charge density. Mg^{2+} ions are commonly used in many DNA-based nanomaterials such as DNA origami for this reason. Regarding the consistency of Mg^{2+} concentration used, we have ensured all data presented in the manuscript now use the same Mg^{2+} concentration. The TEM has been removed from the manuscript due to the new cryo-EM data.

Further, we have performed variable temperature fluorescence measurements in the absence of magnesium ions and we did not observe an assembly transition upon heating (**Figure S5**). This highlights the importance of counterions to screen repulsion.

Please add CLSM images showing higher contrast, the background fluorescence should be seen. In addition, many fibers should be in view not just isolated ones. This is required to help understand the different populations.

We thank the reviewers for their comments. This imaging was performed using total internal reflection fluorescence microscopy – as such, the background is minimized by illuminating the sample at an angle greater than the critical angle, meaning only a thin slice above the coverslip (~ 100 nm) is excited.

Additionally, because these fibers are composed of massive numbers of individual, fluorescently-labelled DNA strands, we performed the imaging at low power (0.7 mW as measured from the objective) and low gain, to not saturate the pixels. As a result, the fibers are so much brighter than the background in these imaging conditions that single fluorophores that are non-specifically attached to the surface are not visible.

To illustrate this, please see the additional images added to the SI (**Figure S9**), where we have increased the brightness and decreased the contrast after imaging. This discussion was added to the figure legend of **Figure S9**.

Figure S9: Total internal reflection fluorescence microscopy images of 9 μM of **bC8₈-DNAa** and 1 μM of **bC8₈-Cy3-DNAa** mixed in 0.5 X TAMg (6.25 mM Mg^{2+}) then heated at a rate of 1 $^{\circ}\text{C}$ per minute to 75 $^{\circ}\text{C}$ and held for 6 hours before cooling to room temperature. Surface was functionalized with a complementary DNA strand. **A.** Additional image. **B.** Additional close-up images of individual fibers. **C.** Demonstration of low background fluorescence. Image on right has increased brightness and lowered contrast.

SI should include side-by-side simulations of all the different disks. Currently its only 2 types.

Unfortunately, we did not have the resources to perform further MD simulation experiments due to changes in circumstances with our collaborators. However, the additional structural evidence provided by cryo-EM strongly supports our model of folding for the DNA-curlamers. The results show that the fibers have a core with dimensions that match our model, and that the fibers formed by **bC8₈-DNAa** are more highly ordered and slightly narrower than those formed by **bC8₁₀-DNAa**.

Line 103 – heated not heating. What does holding for 6 hours mean, at 75 degrees? Don't you need to heat above this temperature to get the fibers to form (line 96)?

We have greatly clarified the heat-activated assembly process throughout the manuscript. "Holding for 6 hours" meant keeping the temperature constant at 75 degrees for six hours, before cooling to room

temperature and imaging. In direct reference to the reviewer's comment, we have included the following in the text:

“Very slow heating (at 0.1 °C / min) to 75 °C (below the apparent assembly temperature of 88 °C) and holding this temperature for 6 hours gave very long and flexible fibers of over 30 μm in length, as observed by total internal reflection fluorescence microscopy (TIRF-M) (Fig. 1D, S9). This assembly at lower temperatures than previously indicated by AFM or fluorescence suggested a high sensitivity to the heating rate. A heat-activated nucleation-growth mechanism could explain these observations, in which nucleation rate gradually increases with temperature. A slower heating rate could result in a lower assembly temperature by facilitating the growth of only a few nuclei into very long fibers. This would be the opposite of the usual mechanism for supramolecular polymerization, where cooling (rather than heating) rate is used to control the assembly^{9,43,46}.”

And to reiterate, the assembly occurs during heating (not during cooling following heating) as demonstrated by variable temperature fluorescence and atomic force microscopy. We propose that heating makes the conformation of the oligomers more dynamic, allowing them to assume the curled state. Once assuming this state, they may nucleate and elongate. In other words, the equilibrium between curled and collapsed conformations shifts towards curled as temperature increases.

Line 191 Fibers stopped growing after 60 mins. Yet in Figure S13 LN numbers are increasing up to 120 mins.

The differences in L_N for samples at 60, 90, 120, 180 mins are not statistically significant, hence our statement that the growth stops after 60 minutes. This remark was added to the figure legend of **Figure S27** in the SI. Also, to clarify, the 60 minute sample is shown in **Figure 4B** within the manuscript.

The curlamer has to be heated above 75 degrees to unfold the hydrophobic block. But the seed growth goes against this and states fibers can form below the critical temperature. Can you explain this?

We have clarified this point in the manuscript:

*“Pre-assembled, short cylindrical particles of **bC8₁₀-DNAa** were added to a solution of **bC8₈-Cy3-DNAb** at a molar ratio of 1 to 10, then heated at 0.5 °C per minute from 25 to 99 °C while monitoring fluorescence. A transition was observed between 70 and 90 °C, lower than the transition observed for **bC8₈-Cy3-DNAa** alone (Fig. S26). In nucleation-growth systems, the energy barrier to self-assembly is typically nucleation rather than elongation; therefore, the lower transition temperature suggested that **bC8₁₀-DNAa** could be acting as seeds to nucleate the assembly of **bC8₈-Cy3-DNAb**. Seeds of **bC8₁₀-DNAa** were again added to **bC8₈-Cy3-DNAb** at a ratio of 1 to 20 and then directly heated at 70 °C (Fig. 4A). An elongation temperature of 70 °C – the very lower edge of the transition observed by fluorescence – was chosen to avoid self-nucleation of **bC8₈-DNAb**. Indeed, nanofibers were observed with very low dispersity in length and grew progressively longer for up to 60 mins, as indicated by AFM (Fig. 4B, C, S27-28). The observed progressive elongation following longer hold times at 70 °C again indicated the unique ability of this system to actively assemble at high temperatures. The resultant fibers exhibited low dispersity in length ($L_N = 790$ nm, $D = 1.02$, $N = 107$ – observed by AFM), consistent with a seeded-growth mechanism of polymerization.”*

To reiterate – the added seeds are lowering the energy barrier to fiber growth during heating, allowing the polymerization to occur at lower temperatures than in the non-seeded experiment.

The authors should show hybridization to assembled DNA fibers using CLSM. This will help prove the orientation and demonstrate the single strand sequence is accessible.

As stated above in response to one of the Reviewer #1's previous comments, we decided to focus on the novelty of using a small molecule to interact with the DNA rather than a complementary DNA strand. In the future, we intend to further explore the potential of attaching other cargo to the fibers using complementary DNA for biological applications.

Figure S4 needs a control – Cy3 might be degrading at 90 degrees over extended periods of time.

As stated above, we have performed additional variable temperature fluorescence studies, with a control in the absence of Mg^{2+} . Fluorescence decreases with temperature (as is expected due to enhanced nonradiative decay rates of the excited electrons), however a clear transition is observed in the case with Mg^{2+} that corresponds to the AFM data presented in **Figure 1**.

Please describe in more detail how the AFM length measurements were collected manually.

The following description in the SI has been edited to clarify:

“Fiber lengths were measured manually using Image J software. Segmented lines were used to trace the fiber paths by hand. Every assembly within an AFM image was measured to remove measurement bias.”

As stated, the fiber lengths were measured by tracing their paths by hand.

Figure S5 and S9 DLS graphs should be the averages of multiple independent repeats, not just technical repeats.

See **Table S1** for the average data. These are from independent experiments.

SI AFM scale bars are missing or not defined properly.

AFM images in the SI have been amended to show either a defined scale bar or the total image width below the micrographs.

SI Samples were purified using HPLC, 20-100 uL was injected with an OD 0.5. This seems really low?

We only had access to an analytical scale column. Multiple runs were performed to purify sufficient material to run the experiments.

How do you calculate the extinction coefficients for each construct?

The extinction coefficients of each specific DNA-oligomer used were estimated by calculating the weighted averages of the extinction coefficients of the nucleotides in the sequences. This has been added to the SI (**SI-II-a**).

Reviewer #3 (Remarks to the Author):

The manuscript from the Sleiman laboratory furthers their understanding and exploration of the assembly of diblock copolymer-like molecules designed with one 'block' being a short, single-stranded DNA segment and the other 'block' being a short phosphodiester backbone with alkyl chain side groups of various lengths. The primary new understanding in this manuscript over their recent publications is the ability to control the length and polydispersity of chains formed via heating of dissolved molecules as well as via seeding of desired assemblies. The results and ability to control chain formation, produce segmented chains (from recent papers), and control dispersity reminds me of the Crystallization Driven Self-Assembly that people use in the literature to control micelle-like nanoparticle formation with essentially living ends that can be added to with addition of new blocky molecules. Since the chains discussed herein require heating (and multivalent cations) to form, the mechanism underlying this assembly seems to be more aligned with something similar to an LCST (lower critical solution temperature) transition or behavior observed in cold-denaturing proteins. But, the living-like assembly and control of the length and dispersity is similar in effect.

I typically find novelty arguments in reviews to be lazy responses to submitted manuscripts. However, in this case, the submission to Nature Communications begs the question-beyond the demonstration of more control of length/dispersity of chain formation, what is the novelty within the work beyond the 2022 JACS communication and 2021 Chem paper? More importantly, however, is the work needed to more definitively understand the nanostructure underlying the chains that are clearly formed with Mg²⁺ and higher temperatures. This, I will dive into in more detail below:

In general, the AFM data in the manuscript seems to undermine the model of assembly proposed of the curling of individual phosphoesters into disks that then stack to form the core of a cylinder that they have a dilute corona of DNA extending from the highly negatively charged phosphodiester backbone. In looking at the diameters and heights of the chains in AFM, the following becomes apparent. In figure 2, bC8-10 and bC8-12 seem to have similar diameters from AFM. This is expected since the change in disk diameter by adding two more backbone units would be quite small. However, bC12-10 and bC12-12 appear to have quite different cross-sections with the shorter molecule bC12-10 having much larger diameter. Even assuming typical AFM tip convolution effects, the differences in diameters between these two molecules' chains is clear, implying a different cross-section or different assembly mechanism since bC12-12 should have the larger cross-section than bC12-10.

Also in Figure 2, the bC8 AFM data suggests something different than what is described in the manuscript. The bC8-12 "spheres" in 2A appear to be very short segments of some kind of similar chain assembly as seen in the bC8-10 and -8, just very short segments of the chains. At the very least, many of the observed nanoparticles are anisotropic in morphology. Similarly, the "spheres" in bC12-8 molecules, while much smaller in cross-section than the chain assemblies and obviously a different morphology, appear to regularly have cavities/holes within the nanoparticles. Perhaps this is due to drying the particles for imaging. However, no similar cavities appear in the chains.

The TEM in 1C is not too helpful in telling the reader anything beyond what the AFM data provides. The drying of the thin films of fibrillar structures displays a hierarchy of diameters/lengths while the AFM reveals individual chains that can be analyzed. If one were to include electron microscopy that could help elucidate what the local nanostructure is, then an in situ imaging technique like cryoTEM would be an excellent complement to the AFM data.

Even with the AFM data, more analysis could be performed in order to better understand the local structure of the 1-D, fibrillar assemblies. For example, some basic line tracing of the nanostructure could provide a more quantitative comparison of the nanostructure and, particularly through the heights measured, could shed light on a different assembly mechanism. For example, it seems through heights measured and diameters that the cross-section of the fibrils is much larger than what would be expected for the proposed disk assembly. This may lead to proposal of a thermodynamically similar assembly mechanisms (e.g., LCST-like collapse of the hydrophobic alkyl chains causing aggregation) but with a much different core (e.g.

extended phosphodiester blocks or chain folded blocks in the core domain). These type of mechanisms/model would allow for more molecules to reside together along the length of the chain providing both a larger cross-section as well as more DNA chains extending into solution as corona chains. The lengthscales/sizes of fibrillar cross-section with the proposed disk core are much too small relative to the observed data. Data from previous papers (e.g., the > 3 nm diffraction-like peak in x-ray scattering) supports the point that there is some longer lengthscale regular packing of longer molecules in the cores and not a single molecule hydrophobic disk that is both very small in diameter as well as in length along the cylinder fibril. The melamine stabilization of fibers would presumably provide the stability needed so that in-depth study of nanostructure/cross-section could take place so that fibril morphology is thoroughly understood. Additional support for a different packing mechanism in the core of the nanostructure is the fact that the authors observe the following when comparing bC8 and bC12. As stated, "This trend from fibers to spheres as the number of bC12 monomers 148 decreased from 12 to 8 is the opposite to that observed for bC8, where spheres are formed with 12 149 monomers and the fibers with 8. Evidently, the hydrophobic to hydrophilic ratio alone is not 150 governing the assembly morphology." Rather than simply state the fact that this puzzle exists, it would be more illuminating for the reader for the authors to propose a packing model that helps describe the actual data and nanostructures and their relative sizes/geometries and not to fall back on the previously proposed cylinder core due to the "curlamer" assembly.

Another aspect of the assembly requires additional thought and will also help with the comments made above about the structure underlying the fibrillar assembly. In the 2021 Chem paper (reference #40), it was shown that bC12-12 shows no assembly without Mg²⁺ (or Ca²⁺) present. Obviously, Mg²⁺ is an important aspect of the assembly in the current manuscript. Unlike the discussion in the other manuscript that only described possible "screening" of electrostatic charges in the phosphodiester block backbone, it is likely the multivalent cations are acting as some type of physical crosslinking agent between the multiple negative charges and acting as an active part of the nanostructure formation in the fibril formation. This would be more favorable at low temperature, but, the fact that assembly into fibrillar structure does not occur both at low temperatures as well as with a lack of divalent cation strongly suggests these ions act as ionic crosslinkers to the phosphodiester backbone charged groups. A simple experiment would be to perform assembly with monovalent salts at similar concentrations and/or ionic strengths and then observe if fibrillar assembly occurs. If it does, then charge screening is primarily the roll of the ions. If not, then there is direct evidence of the importance of the divalent ions in the mechanism of assembly. My hypothesis is that monovalent salts provide a completely different assembly route (if one exists at all). For reference, there are many papers in the literature that describe the complexation of polyelectrolytes with oppositely charged, multivalent ions to form coacervates, crystals, micelles, hydrogel network, collapsed brushes at interfaces, amongst others.

As described earlier, the control of length of fibers with seeding/temperature is a bit incremental when the assembly process has been thoroughly discussed in manuscripts over the past year or more. Perhaps if combined with more nanostructure characterization (e.g. higher rez/mag AFM with line traces, sharper tips to quantify diameter and height, TEM (or cryoTEM) revealing in situ diameters, SAXS of dilute samples using cylinder or cylinder micelle models vs other geometry of cross-nanostructure cross-section for fitting of fibrillar diameter) and a refinement of the model for how molecules are packing within the fibrillar nanostructure, there would be a solid foundation for publication. The simulations performed on individual molecules that show collapse of the hydrophobic side chains at higher simulation temperature is in support of the temperature-induced, LCST-like assembly. However, the purposeful packing of the molecules together into a cylinder core of individual disks is but one version of nanostructure that sequesters the alkyl chains in the core. More straightforwardly, the simulated nanostructure is much smaller in cross-section than the chains observed via AFM. Prior to publication, more thought and analysis should be brought to bear on the nanostructure in order to proposed a more physically meaningful and accurate nanostructure that is supported by the experimental data.

We thank the reviewer for their detailed reading and discussion of the manuscript, which has been helpful throughout the revision process. Below we will respond to the major questions asked by the reviewer. Briefly, the major focuses of our revisions have been:

1. Emphasizing the novelty and general utility of the heat-activated seeded growth mechanism, and performing additional fluorescence and AFM experiments to support it.
2. Elucidating a more detailed model of the nanofiber structure using cryo-EM and 2D particle analysis.

“...what is the novelty within the work beyond the 2022 JACS communication and 2021 Chem paper?”

As we have mentioned above in our answers to the other two reviewers, we acknowledge that the novel points of the manuscript were not adequately emphasized in the initial submission. The most important point of novelty is the heat-activated seeded growth mechanism, which to the best of our knowledge has not been previously demonstrated. The significance of this phenomenon has been stressed through the addition of the following text to the introduction:

“These exemplar supramolecular polymers are soft, water-soluble and dynamic: they require a stimulus to grow, and their length is controlled on demand. Synthetic systems that realise these properties may drive major advances in biotechnologies, ranging from tissue regeneration and biologic manufacturing to rewritable information storage and nanomaterials organization¹⁻³. In addition, many natural supramolecular fibers, such as collagen, are entropically controlled – assembly occurs above a floor temperature⁴. Synthetic supramolecular polymers typically melt at high temperatures and form again upon cooling, due to their dependence on specific intermolecular interactions such as hydrogen bonding. Reversing this paradigm would enable the creation of unique supramolecular materials with functions activated by heat.”

Although our previous work (particularly in *Chem*) shows that heat drives elongation of the fibers, short fibers were always present at room temperature. The system did not undergo a true unassembled to assembled transition with heat as is shown here. We have performed more rigorous variable temperature fluorescence and AFM experiments to further support this, and our results support the previous findings. In addition, this transition from unassembled to assembled allows the seeding of the system to control the fiber dimensions with high precision. As the reviewer mentioned, the results remind them of those obtained by crystallization driven self-assembly; however, the mechanism presented here is driven by conformational changes rather than enthalpically controlled crystallization. It resembles the dynamic systems seen in biology, composed of proteins in aqueous solution.

A second important and unexpected finding in this manuscript is that the self-assembly mechanism strays from the established norms of microphase separation. With an increase in the hydrophobic component, the fiber morphology does not gain stability; instead, it becomes less stable. This surprising outcome is supported by cryo-EM and modeling studies, highlighting the importance of the hydrophobic oligomer's precise sequence and length. Our study examines a previously unexplored area of polymer chemistry: the impact of monomer sequence and exact count on the self-assembly of sequence-defined polymers.

“Even assuming typical AFM tip convolution effects, the differences in diameters between these two molecules' chains is clear, implying a different cross-section or different assembly mechanism since bC12-12 should have the larger cross-section than bC12-10.”

As the reviewer mentions, tip convolution effects make the measurement of widths using AFM inaccurate. The AFM tip is a minimum of 2 nm diameter, resulting in at least an error of +4 nm. Differences in tip quality and age add to this. We performed additional AFM image analysis relying on height measurements, which are more accurate and precise to compare the dimensions by AFM. These are summarized in **Tables S1** and **S2**. **bC12₁₂-DNAa** is not listed as it has been previously published. As the reviewer noted, **bC12₁₂-DNAa** does have a smaller height (5.5 ± 0.7 nm) than **bC12₁₀-DNAa** (8.5 ± 0.6). A similar effect is seen with the new AFM data when comparing **bC8₁₀-** and **bC8₁₂-DNAa**, where two populations are observed for **bC8₁₂-DNAa** with different heights (**Figure 2A, S24, S25**). As described above, the new AFM data utilizes a slower heating rate, allowing the assemblies to assemble

more slowly and avoid kinetic traps. Our explanation for this discrepancy is stated in the figure legend to Figure S25:

“For **bC8₁₂-DNAa, two populations with different heights were observed: short fibers with a height of 5.66 (\pm 0.30) nm and very short, higher cylinders at 9.03 (\pm 0.53) nm. The greater height of the very short cylinders and their inability to elongate into fibers suggests a more dense, globular conformation of the oligomers. The lower height of the other short fibers could indicate a structure in which the branched oligomeric core assumes a more helical conformation to maintain a smaller core diameter, resulting in a lower density of DNA in the corona and hence a lower height by AFM. The short length of these fibers suggests that this conformation is less conducive to forming long fibers compared to **bC8₈-DNAa**.”**

“...then an *in situ* imaging technique like cryoTEM would be an excellent complement to the AFM data.”

We agree with the reviewer that cryo-EM would greatly help us understand the nanoscale structure of the fibers, hence it was carried out on **bC8₈-DNAa** and **bC8₁₀-DNAa** (Figure 2B, 2C, 3C). From 2D particle analysis, the results clearly support our hypothesis regarding the curled conformation of the oligomer backbone. A detailed discussion has been added to the manuscript. Evidence suggesting a slight helical nature to the curled backbone was also gathered, leading us to construct a more detailed model of the fiber structure and further supporting the less ideal geometry of **bC8₁₀-DNAa** compared to **bC8₈-DNAa** (Figure 3C).

Rather than simply state the fact that this puzzle exists, it would be more illuminating for the reader for the authors to propose a packing model that helps describe the actual data and nanostructures and their relative sizes/geometries and not to fall back on the previously proposed cylinder core due to the “curlamer” assembly.

As mentioned above, based on the additional insight gained from cryo-EM, a more detailed model of the fiber structure has been proposed and presented in the manuscript (Figure 3C).

If it does, then charge screening is primarily the roll of the ions. If not, then there is direct evidence of the importance of the divalent ions in the mechanism of assembly.

We agree with the reviewer that the magnesium ions are important for the assembly. Magnesium was chosen because of its biological relevance for DNA stability and its widespread use in the field of DNA nanotechnology⁵. We performed an additional variable temperature fluorescence experiment in the absence of magnesium ions, showing that no assembly transition was observed (Figure S8). However, we do not think that running the experiment with a monovalent cation is useful as a control, as charge screening is not just a factor of ionic strength, but also charge density – monovalent ions screen charge less efficiently than divalent. Furthermore, evidence in the literature suggests that Mg²⁺ ions interact with phosphates through outer sphere coordination, meaning water is present between the two ions⁶. Therefore, it would be unlikely that an ionic bond is forming. Regardless, it is likely that different ions would impact the assembly and will be the basis of future work within the group.

However, the purposeful packing of the molecules together into a cylinder core of individual disks is but one version of nanostructure that sequesters the alkyl chains in the core. More straightforwardly, the simulated nanostructure is much smaller in cross-section than the chains observed via AFM. Prior to publication, more thought and analysis should be brought to bear on the nanostructure in order to proposed a more physically meaningful and accurate nanostructure that is supported by the experimental data.

As described above, we believe we have made significant steps to clarify the molecular packing of the nanostructures, using cryo-EM and AFM analysis. Also, as mentioned, the AFM width dimensions are an inaccurate way of defining fiber diameter. The cryo-EM results support our model of a curled oligomeric core.

References

- 1 Álvarez, Z. *et al.* Bioactive scaffolds with enhanced supramolecular motion promote recovery from spinal cord injury. *Science* **374**, 848-856 (2021). <https://doi.org:10.1126/science.abh3602>
- 2 Aliprandi, A., Mauro, M. & De Cola, L. Controlling and imaging biomimetic self-assembly. *Nat Chem* **8**, 10-15 (2016). <https://doi.org:10.1038/nchem.2383>
- 3 Woods, D. *et al.* Diverse and robust molecular algorithms using reprogrammable DNA self-assembly. *Nature* **567**, 366-372 (2019). <https://doi.org:10.1038/s41586-019-1014-9>
- 4 De Greef, T. *et al.* Supramolecular Polymerization. *Chemical Reviews* **109**, 5687-5754 (2009). <https://doi.org:10.1021/cr900181u>
- 5 Kielar, C. *et al.* On the stability of DNA origami nanostructures in low-magnesium buffers. *Angewandte Chemie* **130**, 9614-9618 (2018).
- 6 Zhou, W., Saran, R. & Liu, J. Metal sensing by DNA. *Chemical reviews* **117**, 8272-8325 (2017).

Reviewers' Comments:

Reviewer #1:

Remarks to the Author:

Interestingly, all referees voiced the same concerns about incremental novelty of this work, which leaves the question open if Nature Communications is the most appropriate outlet. But as I pointed out before, this is in the end an editorial decision.

The authors have thoroughly revised their manuscript and I think it is ready for publication.

Reviewer #2:

Remarks to the Author:

The manuscript has been improved significantly and most points addressed.

I believe Figure 1D is a cluster of nanofibers rather than single fibers - similar to Small Burns 2021 or ACS nano Zhang et al. 2023 publications. Potentially the background objects in the TIRF microscope image are single fibers which have bundled up.

Reviewer #3:

Remarks to the Author:

The authors have made significant revisions of the manuscript in light of my review (as well as the other reviews). All three reviewers discussed novelty with the authors much more clearly pointing out what is new in the current manuscript. If the editors are fine with the novelty discussion, I am pleased with the effort of the authors in revising the manuscript significantly in light of the reviews.

I gave reference 5 in the response letter a look with respect to Mg⁺² effects on DNA and DNA origami structures. This led me down a bit of a mono- vs di- vs trivalent/multivalent cation rabbit hole with respect to effects on DNA (in comparison with or in conjunction with DNA condensing with spermine/spermidine/histones). Since I am not immersed in the DNA origami field, I may simply be using language a bit differently than what is invoked when discussing DNA in particular. My only point about the Mg⁺² is that it is likely condensed/complexed with the DNA as opposed to loosely associated with it like a monovalent ion would be. I was thinking the condensation (whether there is water also present or not) could help lead to conformational changes in the DNA instead of just complexation of opposite charges. Looking at the literature it seems Mg⁺², while complexing with DNA, is not sufficient to cause collapse/assembly of the DNA (as opposed to what is observed with trivalent cations/multiamines/histones). Perhaps a semantic point is that in the general polymer physics community, one wouldn't call that "screening" but rather would describe it as a complexation with screening occurring with soluble, loosely associated ions (no matter the valency). But, the important point is that Mg⁺² helps stabilize origami by complexing out charge of the DNA chain (as opposed to screening it while in solution). Not important here-I think the manuscript is solid and is worthy of publication.

REVIEWERS' COMMENTS

Reviewers' comments are depicted in *italic*.

Reviewer #1 (Remarks to the Author):

Interestingly, all referees voiced the same concerns about incremental novelty of this work, which leaves the question open if Nature Communications is the most appropriate outlet. But as I pointed out before, this is in the end an editorial decision.

The authors have thoroughly revised their manuscript and I think it is ready for publication.

We thank the reviewer for their comments.

Reviewer #2 (Remarks to the Author):

The manuscript has been improved significantly and most points addressed.

I believe Figure 1D is a cluster of nanofibers rather than single fibers - similar to Small Burns 2021 or ACS nano Zhang et al. 2023 publications. Potentially the background objects in the TIRF microscope image are single fibers which have bundled up.

The reviewer brings up an interesting point regarding the possibility that the fibers are clustered together. As stated in the methods section, the DNA fibers are diluted 2000X prior to imaging, and as such we would not expect to see a dense carpet of fibers on the glass coverslip. Figure 1D also does not show any branching, which would be expected if many fibers were clustered or bundled together. Finally, the fiber length in Figure 1D is similar to some of those seen in Supporting Figure 1 by AFM, further supporting our claim that individual fibers are being observed by TIRF.

Reviewer #3 (Remarks to the Author):

The authors have made significant revisions of the manuscript in light of my review (as well as the other reviews). All three reviewers discussed novelty with the authors much more clearly pointing out what is new in the current manuscript. If the editors are fine with the novelty discussion, I am pleased with the effort of the authors in revising the manuscript significantly in light of the reviews.

I gave reference 5 in the response letter a look with respect to Mg²⁺ effects on DNA and DNA origami structures. This led me down a bit of a mono- vs di- vs trivalent/multivalent cation rabbit hole with respect to effects on DNA (in comparison with or in conjunction with DNA condensing with spermine/spermidine/histones). Since I am not immersed in the DNA origami field, I may simply be using language a bit differently than what is invoked when discussing DNA in particular. My only point about the Mg²⁺ is that it is likely condensed/complexed with the DNA as opposed to loosely associated with it like a monovalent ion would be. I was thinking the condensation (whether there is water also present or not) could help lead to conformational changes in the DNA instead of just complexation of opposite charges. Looking at the literature it seems Mg²⁺, while complexing with DNA, is not sufficient to cause collapse/assembly of the DNA (as opposed to what is observed with trivalent cations/multiamines/histones). Perhaps a semantic point is that in the general polymer physics community, one wouldn't call that "screening" but rather would describe it as a complexation with screening occurring with soluble, loosely associated ions (no matter the valency). But, the important point is that Mg²⁺ helps stabilize origami by complexing out charge of the DNA chain (as opposed to screening

it while in solution). Not important here-I think the manuscript is solid and is worthy of publication.

We thank the reviewer for their appreciative comments about the revised manuscript. We agree with the reviewer's points regarding the likely relatively close interaction between Mg^{2+} and the DNA, which stabilizes the fibers.